# CFR-YOLO: A Novel Cow Face Detection Network Based on YOLOv7 Improvement

**DOI:** 10.3390/s25041084

**Published:** 2025-02-11

**Authors:** Guohong Gao, Yuxin Ma, Jianping Wang, Zhiyu Li, Yan Wang, Haofan Bai

**Affiliations:** School of Computer Science and Technology, Henan Institute of Science and Technology, Xinxiang 453003, China; mayuxin@stu.hist.edu.cn (Y.M.); wangjianping@hist.edu.cn (J.W.); lizhiyu@stu.hist.edu.cn (Z.L.); wangyan@stu.hist.edu.cn (Y.W.); baihaofan@stu.hist.edu.cn (H.B.)

**Keywords:** cow face detection, YOLOv7, deep learning, target detection

## Abstract

With the rapid development of machine learning and deep learning technology, cow face detection technology has achieved remarkable results. Traditional contact cattle identification methods are costly; are easy to lose and tamper with; and can lead to a series of security problems, such as untimely disease prevention and control, incorrect traceability of cattle products, and fraudulent insurance claims. In order to solve these problems, this study explores the application of cattle face detection technology in cattle individual detection to improve the accuracy of detection, an approach that is particularly important in smart animal husbandry and animal behavior analysis. In this paper, we propose a novel cow face detection network based on YOLOv7 improvement, named CFR-YOLO. First of all, the method of extracting the features of a cow’s face (including nose, eye corner, and mouth corner) is constructed. Then, we calculate the frame center of gravity and frame size based on these feature points to design the cow face detection CFR-YOLO network model. To optimize the performance of the model, the activation function of FReLU is used instead of the original SiLU activation function, and the CBS module is replaced by the CBF module. The RFB module is introduced in the backbone network; and in the head layer, the CBAM convolutional attention module is introduced. The performance of CFR-YOLO is compared with other mainstream deep learning models (including YOLOv7, YOLOv5, YOLOv4, and SSD) on a self-built cow face dataset. Experiments indicate that the CFR-YOLO model achieves 98.46% accuracy (precision), 97.21% recall (recall), and 96.27% average accuracy (mAP), proving its excellent performance in the field of cow face detection. In addition, comparative analyses with the other four methods show that CFR-YOLO exhibits faster convergence speed while ensuring the same detection accuracy; and its detection accuracy is higher under the condition of the same model convergence speed. These results will be helpful to further develop the cattle identification technique.

## 1. Introduction

Along with the implementation of the policy of strengthening agriculture, the proportion of livestock farming in China’s agricultural output has been increasing. Beef cattle breeding in the traditional livestock breeding system occupies a large proportion, and in the process of beef cattle breeding, we need to consider the individual differences between the cattle and the different growth cycles of the cattle for the targeted development of breeding programs to maximize the profitability of farming [1]. However, in recent years, the uncertainty in the cattle breeding process has risen under the influence of factors such as epidemic prevention, environmental protection, and the high volatility of agricultural markets. In order to reduce the risk of farming, many cattle enterprises have put forward the concept of precision livestock farming. The accurate identification of cattle individuals is the core of accurate cattle breeding, and at the same time, in view of the current development trend and market demand, in combination with the cattle breeding industry, the cattle individual identity information monitoring is increasingly developing in the direction toward becoming non-destructive, high-accuracy, intelligent, and automated [2]. Therefore, in order to realize precision animal husbandry, it is of great significance to establish a contactless, high-precision, and highly automatized process for the individual identification of bovine animals.

The traditional cattle individual identification is mainly marked by physical methods to achieve the identification purpose by manually identifying the facial features of the cattle, including ear marking method, ear truncation method, cattle horn branding method, stabbing ink method, embedded chip method, etc. [3]. These identifiers can be tracked using RFID (radio frequency identification technology) or GPS systems, and they can be used to manage herds on large ranches. Although this method is highly accurate, invasive physical marking of individual cattle and then identification of cattle by observing facial features makes this process prone to loss and tampering, and animal welfare is also poor. Moreover, physical marking makes it difficult to realize all-day behavior monitoring of cattle, and individual information of cattle cannot be timely fed back to breeding managers. Problems that arise cannot be identified and adjusted in time. In this way, it will influence the productivity increase, and it will not meet the requirement of modern large-scale breeding.

In the cattle breeding industry, the implementation of all-weather video surveillance of cattle has important significance. Video monitoring enables breeders to monitor the animal’s behavior and health status in real time, by means of cattle activity, feeding habits, and interaction with other cattle; can detect signs of disease, injury, or stress behavior in time, so that rapid action can be taken to treat or adjust management strategies; and reduce the need for manual inspection, especially in large-scale farms. It can effectively save labor costs [4]. In addition, the highly automated monitoring system can also record and analyze data to help managers make more scientific and rational farming decisions. The video monitoring system offers a non-invasive method for continuous observation of cattle that greatly improves the welfare of the animals and improves the efficiency of the breeding, ensures the welfare of the animals and optimize the management of the resources. Against this background, how to achieve effective and high-accuracy individual identification of bovine animals, while at the same time promoting animal welfare, optimal resource management, and disease regulation, is a core element of round-the-clock herd monitoring, and the advantages of bovine face detection technology in modern cattle farming perfectly fit this concept.

Bovine face detection is a kind of identification by collecting facial features of cattle using computer vision technology. The application of facial detection technology in animal detection is similar to face detection, where facial images are captured through cameras, and deep learning models, such as convolutional neural networks, are used for feature extraction and matching. This technique distinguishes and identifies different cattle by analyzing unique features of their faces, such as shape, size, color, and pattern distribution [5]. Its advantage is that it does not require physical contact, enabling remote and non-intrusive monitoring. Cattle face detection systems typically take pictures or videos with high-resolution cameras and then process these images through specific algorithms to achieve accurate identification of each cow [6]. The main advantage of this technology is that it is non-invasive and highly automated, allowing continuous and precise identification of an individual without disturbing the animal’s natural behavior.

In contrast to the traditional method of contact identification, the non-contact cattle face detection is based on the unique and stable biometric image of the cow. It does not need to rely on external intervention, and it completely uses the biological characteristics of the cow as a marker to achieve individual identification. Biometric features are not easy to be copied, tampered with, or lost, and the cost of image acquisition is more economical than that of physical labeling. It not only enhances the function of video surveillance but also associates behavioral data with specific individuals by accurately identifying the identity of each cow, providing more accurate data support for research [7]. Cow face detection technology faces multiple challenges in current practical applications, including light variations, occlusion problems, multiple cow faces, and the need for high-performance algorithms and computational resources [8]. These issues affect the accuracy and efficiency of detection systems, especially in large-scale cattle-farming environments. Unstable lighting conditions may lead to difficulties in image capture, while occlusion further reduces the robustness of the system. In addition, complex algorithms require significant computational resources, thus increasing the complexity of the technology implementation.

Therefore, in order to improve the application effect of bovine face detection technology, this paper proposes CFR-YOLO, an improved network model for bovine face detection in large pastures based on YOLOv7. Its main contributions are as follows:Based on the features of the cow’s face (nose, mouth, and eye corners), a method of extracting the features of a cow’s face is constructed. Calculate the center of mass and frame for the nose, mouth, and eye corners of beef cattle.An improved bovine face detection method based on YOLOv7 was designed. Specific optimizations include replacing CIoU loss functions with SIoU loss functions. FReLU activation function is used to replace SiLU activation function: the CBS module is changed to a CBF module. Introduce the RFB module into the backbone network. The convolutional block attention module (CBAM) is introduced in the head layer to optimize the CFR-YOLO model.The performance of the CFR-YOLO model is evaluated by experiments on self-built datasets. Compared with existing methods, such as YOLOv7, YOLOv5, YOLOv4, and SSD, the advantages of the proposed method in bovine face detection tasks are verified.

This article comprises six sections: Section 2 is a brief overview of the development of cow face detection methods and an introduction to the current mainstream detection methods; Section 3 outlines the foundational architecture of YOLOv7 and its enhancements; Section 4 details the materials and methods employed; Section 5 discusses the experimental analysis; and Section 6 concludes the study.

## 2. Status of Research

### 2.1. Status of Traditional Cow Face Detection Technology

Currently, in traditional identification methods, individual cattle are marked through the use of manual methods, such as ear engraving, ear tagging, branding, and neck chaining. The ear-tagging method is not only easy to install and inexpensive, but it is not easy to harm the individual animals when using it, and the managers of the farms can easily recognize the number of the cattle by observing them with their naked eyes. However, this method faces major problems when ear tags are lost, making it ineffective for long-term use [9]. The ear-carving method causes pain to the animals and is not applicable to medium and large farms [10]. Conventional labeling methods are both time-consuming and labor-intensive, hindering automated management in dairy farms. Traditional livestock individual identification can cause some physical damage to the animals, as well as being time-consuming and labor-intensive. With the continuous development of electronic equipment, there are now methods of identification using electronic labeling equipment such as barcode ear tags and radio-frequency identification technology.

Ear-marking identification method: The main methods of identifying individual cattle by ear tags are barcode ear tags and metal ear tags. Barcode ear tags identify animals by placing a unique barcode on the surface of an ordinary plastic ear tag, which enables the querying of animal information and thus the traceability of the animal [11]. However, when using 2D barcodes for animal identification, the method is easily contaminated and damaged, and it has a low life span [12,13]. Metal ear tags are the primary method for individual identification on small-to-medium-sized farms and remain extensively used by ranchers. Ear tags are one of the most widely accepted methods of identifying individual cattle. This method is relatively convenient and cost-effective. Compared to techniques like hot-iron branding, ear tags are less invasive and cause minimal harm to cattle, but wearing ear tags for a long period of time can also result in the rotting of the cattle’s ears, and ear tags are highly susceptible to copying and counterfeiting, and they are easy to lose, which is unfavorable to the development of cattle insurance business.Radio-frequency identification (RFID) technology: As it is not only difficult but also inefficient to identify individual livestock by manual marking alone, have electronic marking based on the RFID principle has been utilized by Adrion and Cappai for the identification of individual livestock and monitoring of farming information in farming management [14,15]. RFID tags can record an animal’s age, sex, breed, and color, and the use of RFID technology in cattle farming management further improves the level of automation in the farming management process. Geng et al. [16] used radio-frequency technology for cow identification to manage the cattle in the farm. Xiong and Sun et al. [17] designed a milk yield measurement system based on RFID technology and image processing. This approach ensured system accuracy, enhanced stability, and lowered production costs.

While traditional methods of individual cattle identification, such as ear tags and neck collars, provided society with a basic identification tool in the early days, these methods no longer meet the growing demands of modern animal husbandry for animal welfare and efficient precision farming. These traditional methods not only fail to achieve real-time monitoring of cattle around the clock, but they also require a large amount of manpower and resources to manage, and there is a risk of causing stress and physical injury to the animals, which are particularly prominent in large-scale pasture management. In addition, the vulnerability of ear tags and neck loops to damage and dislodgement can affect the reliability of long-term tracking. With the cattle breeding industry shifting toward large-scale and intelligent practices, the limitations of traditional identification methods have increasingly become a bottleneck hindering industry progress.

### 2.2. Vision-Based Cow Face Detection Techniques

The background of traditional cattle identification technology not being able to meet the development needs of the industry has given rise to the development of cattle face detection systems based on biometric image technology, which can accurately identify individual cattle without direct contact with the animal through efficient image processing and machine learning algorithms. In large ranches, vision-based cattle face detection technology offers significant advantages because it enables automated and real-time identification and tracking of large numbers of cattle without the need to rely on traditional ear tags or electronic tags, which are prone to damage, loss, or misinterpretation in large-scale management. Visual detection technology not only improves the accuracy and efficiency of detection but also reduces manual intervention and operational complexity, significantly reducing management costs, especially for large ranches that need to process large amounts of data and carry out accurate management. This technology enables continuous real-time monitoring to promptly identify health issues and optimize farm operations.

Cow face features belong to the external features of the animal itself and have the characteristics of easy collection and zero harm to the animal itself. Therefore, there are many scholars who have begun to take the cattle face image as a research object. Song et al. [18] proposed a new method for cow face detection that is based on local binary pattern features using SRC (Sparser Representation Classifier) method. For feature extraction, principal component analysis is applied to derive more effective feature representations. Lv et al. [19] proposed an incremental detection algorithm framework to achieve real-time accurate incremental detection in complex backgrounds by taking Holstein cows as the object and using the characteristics of CNN, such as good discriminability, strong migration ability, fast computation of Sparse Representation Classifier (SRC), and easy-to-append features. Huang et al. [20] improved the base network of SSD model and so on for cow face detection to improve the model detection performance. Yang et al. [21] developed and tested a method to improve detection accuracy and recover identity information to generate cow faces close to real identities. The network structure consists of two parts, which are used to recover high-resolution super-resolution network and bovine face detection network from low-resolution bovine faces, and the experimental results achieved good detection accuracy in small-size bovine faces.

To enhance the efficiency of animal husbandry, a large number of researchers have made valuable efforts to design and develop a robust and automatic bovine face detection system. Xia et al. [22] combined principal component analysis (PCA) and Chi-square distance detection to extract features of bovine face images, and used sparse coding to calculate classification results for the extracted features. However, this method ignores the pose factor of the bull face and only focuses on the positive bull face, which requires a lot of image acquisition work in the early stage and is difficult to apply in practice. Cai et al. [23] used LBP operators to extract cow face image features and then preprocessed the extracted features by sparse and low-rank decomposition, effectively improving the algorithm’s detection performance in the test set. However, the algorithm also ignores the influence of the cow’s face posture. Kim et al. [24] collected a small sample of cow face dataset composed of 12 cows in the actual pasture, used associative memory neural network to learn the cow face detection model, and achieved good performance. However, the computational complexity of associative memory neural network is high, which is not conducive to the application of algorithm deployment. Chen et al. [25] introduced feature bagging into the task of individual cow detection based on self-constructed cow face data. Firstly, the optimized directional gradient histogram is utilized to extract features from the cow face image; after that, the spatial pyramid matching is used in the algorithm to calculate the classification result of the cow face image; and, finally, a high detection rate is obtained in the self-constructed dataset. However, the algorithm uses the rectangular feeling field of SPM for matching, which adds image background noise in the calculation. Li et al. [26] added a compression module and an excitation module for channel attention in Mobile Net to obtain high detection gain in image local occlusion scenarios. However, the algorithm focused solely on channel information correlation while neglecting spatial information correlation through channel attention.

The main problem with traditional methods of individual cattle identification is that they rely on markings or equipment on the livestock rather than the animal itself, whereas biometrics offer a fast and reliable method. In today’s information age, face detection technology has been successfully applied in various industries, and detection based on visual biometrics has become a new development trend in the identification of individual animals. Vision-based cattle face detection technology not only optimizes the data collection and monitoring process; it also improves overall animal welfare by reducing human intervention and reducing animal stress. Moreover, continuous data monitoring aids in the early detection of health issues, supporting precision treatment and management, which is crucial for enhancing production efficiency and lowering disease incidence. Therefore, the vision-based bovine face detection technology has not only become the technical trend of modern aquaculture but also a key driver for advancing higher technology and automation in animal husbandry.

### 2.3. Cow Face Detection Technology Based on Deep Learning and Target Detection

Object detection has long been a key focus in computer vision research, particularly in areas such as pedestrian, vehicle, and ship detection [27]. Bovine face detection is also a form of target detection and can thus be achieved using object detection techniques in computer vision.

The research shows that feature fusion plays an important role in improving the accuracy of target detection. Yang et al. [28] improved the detection accuracy of cows’ faces to 93.68% by integrating coordinate information in YOLOv4. Wang et al. [29] improved YOLOv3, introduced deformable convolution and adaptive spatial feature fusion, and achieved an accuracy of 92.8%, close to Faster-RCNN. The ASFF method proposed by Zhu et al. [30] significantly improves the performance of YOLOv5 by weighted fusion of high-level semantics and low-level spatial features. These studies show that multi-feature fusion can effectively improve the detection accuracy, but its application in beef cattle identity detection is still limited.

The YOLO model has been successfully applied in many poultry animals, and individual identification of domestic animals is mainly carried out for common domestic animals, such as chickens, pigs, cattle, and sheep. Li et al. [31] used the YOLOv4 target detection model to automatically monitor the behavior of chickens in the chicken house and extracted the behavior exceeding the time threshold according to the image time sequence. The overall average accuracy reached 79.69%, although the accuracy for movement behavior was relatively low. In the field of pig individual detection, He et al. [32] introduced Dense Block into Darknet53, the backbone feature extraction network of YOLOv3, and combined it with the undersampling layer to form a new network. The improved Spatial Pyramid Pooling (SPP) was added, and the mAP of the model on the test set was 90.18%. Wei et al. [33] conducted hyperparameter optimization for YOLO v3 target detection algorithm for sheep face detection, taking it as a pre-step in sheep face detection. The accuracy of the model on the test set was higher than 97%. These success stories show that despite the challenges of various animal detection tasks, the YOLO network can effectively meet these challenges with appropriate technical adjustments and optimization. In cow face detection, failure cases are also common. The LAM method adopted by Kim et al. [34] has major problems in real-time performance and fails to meet the needs of real-time monitoring. Yao et al. [35] used SSD, Faster-RCNN, and other methods for bovine face detection, but the actual application scenarios of the model were still limited because individual identification was not involved. The Faster-RCNN method proposed by Gou et al. [36], based on improved NMS, has improved the accuracy and recall rate, but its robustness under complex background still needs to be further improved. These challenges provide valuable directions for future research, which may focus on improving detection speed, dealing with interference in complex contexts, and combining individual identification with detection tasks to further enhance the practical application value of the model.

Meng et al. [37] improved the YOLOv7-Pose network model and collected data at Hua Hao Ecological Farming Co., Ltd., Lu’an City, Anhui Province; and Xuyi Weigang Herding Co., Ltd., Huai’an City, Jiangsu Province. Holstein cows were photographed using a webcam with a resolution of 1920 × 1080 in a natural feeding environment on a cattle farm. A total of 846 images in jpg format were obtained by converting videos to images and filtering them. The dataset covers complex scenarios with different lighting conditions, occlusion, blurring, different viewpoints, and single and multi-targets, which fully reflects the complexity of the cattle farm environment. To improve the generalization ability and robustness of the model, the researchers performed data enhancement on the images, including up-and down-flipping, adding noise and random points, etc., to extend the dataset to 2538 images. The dataset is randomly split into training and test sets at an 8:2 ratio, with the training set comprising 2031 images and the test set comprising 507 images. Qi et al. [38], on the other hand, proposed an improved Yolov7-tiny model focusing on cow face detection in complex scenes. To validate the algorithm’s effectiveness, researchers selected 371 cow images from the COCO dataset and 202 real-world cow images captured at the ranch, totaling 573 images, and annotated them using Labellmg. Due to the small dataset, the researchers added stochastic data enhancement techniques to the YOLO algorithm, including random masking, flipping, scaling, color gamut transformation, contrast adjustment, and Mosaic data enhancement methods, which finally expanded the dataset to 5438 images. Subsequently, 300 images from complex scenes were selected as the test set, while the remaining images were split into training and validation sets at a 9:1 ratio. The two datasets mentioned above differ in focus regarding size, complexity, and applicability. The Huang et al. dataset has more complex scenes and is suitable for multi-target detection, whereas Qi et al.’s dataset focuses on the cow face detection task and uses a variety of enhancement methods to compensate for the lack of dataset size. In YOLOv7 improvement, the data enhancement technique plays a key role in improving the generalization ability and robustness of the model, while the reasonable dataset division ensures the training and validation effect of the model.

### 2.4. Summary

The above methods often result in low detection accuracy when recognizing cow faces because of insufficiently accurate feature extraction or insufficient model complexity. While conventional cattle face detection techniques provide real-time data, there are obvious limitations to this method, such as the potential for causing discomfort in cattle and the difficulty of sustained use over a long period of time, which is contrary to the requirements of animal welfare. Additionally, this method lacks data accuracy and adaptability to cattle. The decision-making process of deep models such as those in the CNN family is opaque and difficult to interpret, which is a limitation in application scenarios that require high interpretability. Although the predecessor YOLO family of models has a significant speed advantage, its accuracy may not be as good as other deep learning models in some complex scenarios. In view of this, this paper proposes a novel cow face detection method called CFR-YOLO, based on the new cow face detection network improved by YOLOv7. Specifically, we extracted the feature key points in the region of bovine nostrils, eye corners, and mouth corners by facial key-point features and processed the features using the CFR-YOLO network architecture. Finally, we achieved high performance detection of the model by optimizing the loss function.

## 3. Bovine Face Detection Method Based on YOLOv7

### 3.1. System Architecture of CFR-YOLO

In this paper, a cow face detection method based on YOLOv7 network to extract key feature points of cow faces, named CFR-YOLO, is proposed. In this method, the captured images and videos are clipped according to the input size of the model, and the image data are obtained as the input of the system. Secondly, the image is fed into the improved backbone network, where features are extracted at various levels using algorithms based on nostril, eye distance, and mouth size. The RFB and CBF modules are employed to expand the model’s receptive field, enhance detection accuracy, and minimize computational delays during training. Three of the feature layers then carry out the next network construction, which is called the effective feature layer. Then, three effective feature layers are sent into the neck network to continue feature extraction. In this process, features are not only upsampled and downsampled to achieve feature fusion, but also enhanced feature extraction is achieved through the introduced CBAM module, so that features of different scales are combined, which is conducive to extracting more comprehensive features. Finally, three effective feature layers of different sizes are sent into the head network after enhanced extraction to obtain the detection results. The SIoU (Structured Intersection Over Union) loss function is used for optimization at the output. The system architecture of the entire CFR-YOLO is shown in Figure 1.

Figure 1 shows the overall architecture of the CFR-YOLO bull face detection system. The system realizes the accurate extraction and detection of bovine facial features by optimizing the combination of several modules. First, the original input image is converted into the input data suitable for the model by image preprocessing. Then, through the improved YOLOv7 backbone network, combined with RFB and CBF modules, the receptive field of the network is enhanced, and the ability to capture bovine facial details is improved. Then, through the feature fusion of the neck network and the guidance of the CBAM module, the network can combine features of different scales and finally output accurate detection results. Finally, the SIoU loss function is used to optimize the model, which further improves the detection accuracy and robustness. The whole system can effectively avoid the calculation delay that may occur in the training process and maintain high detection accuracy.

### 3.2. YOLOv7 Network

The YOLOv7 algorithm, introduced by Alexey Bochkovs-Kiy’s team in 2022, outperforms YOLOv5 in both detection accuracy and speed. The overall structure of YOLOv7 is composed of four parts: input layer, backbone network, head, and prediction terminal. The model structure is shown in Figure 2.

The backbone of YOLOv7 network consists of a CBS module, ELAN module, and MPC-B module. The CBS module includes three parts: convolution, batch regularization layer, and SiLU activation function. The MPC-B module consists of one pooling layer and three CBSs, which are used for downsampling. Simultaneously, combining convolution and pooling layers enables users to capture information from all values within a small local region, avoiding the disadvantage of the pooling layer only obtaining the maximum value. ELAN module is composed of multiple CBSs, which form an efficient and converged network structure. It removes 1 × 1 convolution, improves GPU computing efficiency, and greatly reduces the consumption of access memory. Moreover, it adopts the idea of gradient segmentation, and it directly adds short connections to the output and input layers of the convolutional network, so that the gradient flow propagates in different network structures. The overexpansion of input and gradient information is solved, and the shortest and longest gradient paths can be controlled, so that the network can extract more features and make the training more efficient and accurate. The head layer is mainly composed of an SPPCSPC module, ELAN-H module, MPC-N module, UPSample module, and RepConv module. SPPCSPC is a spatial pyramid pooling improvement module, which is composed of multiple CBS modules and pooling layers. Different sensitivity fields can be obtained through maximum pooling. While enabling the algorithm to adapt to images with different resolutions, it can prevent distortion in the process of image cropping and shrinking, and it can avoid repeated extraction of image features by convolution, thus reducing the amount of computation. I can also accelerate the generation of preselected frames. The ELAN-H module closely resembles the ELAN module in functionality and structure, differing in that ELAN sums three outputs in the second branch, whereas ELAN-H sums the results of five CBSs. The MPC-N module extends the MPC-B module by adding a link to the forward output layer. The UPSample module performs upsampling, using nearest-neighbor interpolation to reduce network computation. The RepConv module is a structural reparameterization module that transforms changes in structure during training into changes in parameters during inference, and equivalent substitutions of structure into equivalent substitutions of parameters, thus improving performance and saving space. The final prediction end includes loss function computation and bounding box prediction.

### 3.3. Improved YOLOv7

To enable the model to capture both local and global information, enhance the accuracy and speed of cow face detection, and minimize misdetection, this paper introduces improvements to the YOLOv7 model. Firstly, in the backbone layer, the SiLU activation function is located in the CBS in the backbone network. The CBS module is replaced by the CBF convolutional module, and the SiLU activation function is replaced by FReLU activation function to avoid calculation delay in training process. Secondly, to expand the model’s receptive field and enhance the algorithm’s detection accuracy, the RFB module is integrated into the 24th layer of the backbone network. Finally, in the head layer, a feature extraction enhancement module, is added. To allow the network to extract more comprehensive and representative feature information, the CBAM convolutional attention module is incorporated. This module assigns feature weights across both channels and spatial dimensions, enhancing the extraction of key features. The improved model structure is illustrated in Figure 3, where the “cat” denotes the concatenation operation.

#### 3.3.1. Hybrid Attention Mechanism CBAM

The attention mechanism is a crucial approach in computer vision for enhancing the effectiveness of object detection. It only focuses on the key areas in the image and ignores other irrelevant information. In cattle farms, due to lighting, occlusion, background, and other factors, the interference information in the picture is confused with the feature information of the cow’s face, which affects the accuracy of detection. When cow detection relies on the face, it is highly dependent on detailed facial features and spatial characteristics. Adding feature extraction enhancement module to the network can make the network extract more adequate and representative feature information. The CBAM module can distribute the weight of features from both channel and space, so as to enhance the extraction of key features, so it is used as the feature extraction enhancement module of the network.

As shown in Figure 4, CBAM attention module is a hybrid attention module that integrates channel and spatial information. The channel attention mechanism (CAM) performs global maximum pooling and global average pooling operations on the input feature graph F, compressing the feature graph of each channel into a C × l × 1 feature graph. The compressed feature map is then sent to the shared multi-layer perceptron (MLP), where the perceptron output features are added element-wise. The channel weights of the feature map are normalized using the Sigmoid activation function. Finally, the input feature map FFF is obtained by multiplying the normalized weights with the input feature map FFF. Spatial attention mechanisms (SAMs) enhance model performance by learning the significance of each spatial location. In the spatial attention module, the input feature graph F is firstly averaged and maximally pooled. The two output feature maps are concatenated along the channel dimension, followed by dimensionality reduction using a 7 × 7 convolution kernel. The spatial attention feature is then generated through a Sigmoid function operation, and finally the spatial attention feature is multiplied with F’. Thus, the feature graph F″, after processing by the convolutional attention module, is obtained. In cow face detection, there is a high similarity between cow faces of different cows, and the original YOLOv7 model cannot extract effective feature information completely in the process of feature extraction, which may cause false detection. The channel attention mechanism effectively manages the distribution relationships among feature map channels, and the spatial attention mechanism makes the model pay more attention to the pixel region where the cow face is located in the cow face image while ignoring the pixel region where the background or occlusion is located. By combining the two, the correlation between the cow face features in different dimensions can be captured to improve the detection effect of the model.

#### 3.3.2. Introduction of FReLU Activation Function

FReLU activation function has some advantages in target detection tasks. For example, on the COCO public dataset, ResNet-50 and ShuffleNetV2 are used as the backbone network for experiments. For ResNet-50 networks, the mAP using FReLU activation function was 36.6%, an increase of 1.4% and 0.8% compared to ReLU and SiLU activation functions, respectively. FReLU activation functions also perform better than ReLU and SiLU activation functions in ShuffleNetV2.

Although the FReLU activation function itself imposes less computational overhead, replacing the activation function throughout the YOLOv7 network will inevitably result in computational delays during training. Based on the characteristics of convolutional neural networks, the shallow layers of the network can extract abundant feature information. Therefore, this paper replaces the SiLU activation function in the original YOLOv7 backbone with the FReLU activation function in this section to enhance the feature extraction capability of the YOLOv7 backbone, so as to improve the detection effect of the whole network. The SiLU activation function is located in the CBS module in the backbone network, making the improved CBS module a CBF module.

The fundamental reason for this improvement is FReLU’s ability to effectively mitigate the “vanishing gradient” problem common to RELUs and their variants, especially in deep neural networks. By introducing a filtering mechanism, FReLU effectively controls the effect of negative values, thus avoiding the situation of gradient explosion or disappearance during training. In addition, FReLU can provide stronger expression when dealing with nonlinear features, allowing the network to capture more complex features and, thus, improving the detection accuracy. In contrast, although SiLU activation function can improve the nonlinear expression ability in some scenarios, its gradient is small in the negative region, and it is affected by the “saturation region” effect in some tasks, resulting in a slow convergence rate during training. In experiments, FReLU, through its more balanced activation characteristics, improves the training efficiency of the network and improves the accuracy, especially in complex backgrounds and changing object attitudes.

However, while FReLU is superior to SiLU in some respects, we still recognize the advantages of CFR-YOLO in object detection. The CFR-YOLO performs well in detection accuracy and robustness by combining multiple optimization modules, especially in refined target detection tasks. Therefore, the introduction of FReLU does not weaken the core advantages of CFR-YOLO; on the contrary, it provides CFR-YOLO with stronger feature expression ability, making the model’s performance in complex scenes further improved.

To sum up, although FReLU has significant advantages in terms of improved performance, the comprehensive optimized design of the CFR-YOLO, especially when dealing with multiple features in target detection, is still the fundamental reason for its superior performance. We believe that the combination of FReLU and CFR-YOLO provides a more efficient and robust solution for the field of target detection.

#### 3.3.3. RFB Module

The main role of the YOLOv7 backbone network is to perform feature extraction on the input image, but the original YOLOv7 backbone network has a relatively small receptive field. This results in excessive local information and insufficient global information, ultimately affecting detection accuracy. To further expand the model’s receptive field and enhance algorithm accuracy, the RFB module is incorporated into the backbone of the original YOLOv7.

The RFB module mimics human vision by linking the receptive field to eccentricity, with a structure resembling that of Inception, which mainly consists of multi-branched convolutional layers composed of regular convolutions at different scales and cavity convolutional layers at different scales. The multi-branch convolutional layer is mainly used to simulate the different sensory fields in the population sensory field, and the null convolutional layer is mainly used to simulate the correlation between the scale of the population sensory field and the eccentricity. This structure not only avoids increasing the number of parameters but also expands the receptive field and enhances the algorithm’s detection accuracy.

Figure 5 shows the structure of the RFB network. The input feature map is firstly passed through three branches to complete the feature fusion. The three branches are then respectively passed through 1 × 1 convolution (Cony) to reduce the number of channels of the feature map, and two of them are then passed through 3 × 3 and 5 × 5 convolution kernels, forming a multi-branch structure to capture multi-scale features, and an expansion rate of 1, 3, and 5 is introduced into each corresponding branch, respectively. A 3 × 3 dilated convolution is applied to each corresponding branch to expand the network’s receptive field. When the dilation rate is l, the cavity convolution is the regular convolution. The size of the sensory field is obtained differently when the cavity rate is different. The RFB module sets the dilation rate to a combination such as 1, 3, or 5, which allows the sampling points to be interleaved to learn as much local information as possible, reducing the loss of information due to the grid effect. Finally, the outputs of the three branches are joined together by a 1 × 1 convolution and splicing operation (Concatenation) to complete the feature fusion. The fused feature maps are tensor-summed with the input feature maps via the Shortcut layer, and the final output is generated using the FReLU activation function.

### 3.4. CFR_YOLO Method Design

(1)Facial key-point feature extraction

Facial key-point extraction is a method of forming skeleton information by detecting target key points and connecting them sequentially. The key feature points of the bovine face include the nostrils, corners of the mouth, and eyes. The original image is selected for feature point labeling, and the labeled regions are divided into nostril region, mouth corner region, and eye corner region. The eye-corner feature points are noted as C1–C2, and based on the coordinate positions starting from the left-eye corner region, the nostril feature points are recorded as B1–B6. The mouth corner feature points are denoted as E1–E2. The feature point representation of the bovine face is shown in Figure 6.

(2)Cow face detection

Frontal, left-side and right-side images of the cow are captured, respectively, and then the captured facial image data of the cow are preprocessed. The outline of the cow’s face in the image is outlined, and the identity of the cow is labeled. The preprocessed cow face image is converted into a JSON (JavaScript Object Notation) file, which contains the coordinates of the cow face region in the image. the cow’s identity information and the path to the directory, where the image is located. Deep learning algorithm is used to extract the data in the cow face part of the JSON file for learning to obtain the cow face feature extraction model. The extracted cow face image features are stored in separate feature vector databases based on the specific conditions. These include the frontal, left-side, and right-side face feature vector databases.

The dataset was split into a training set, validation set, and test set in a 7:2:1 ratio to develop the CFR-YOLO deep learning model for localizing the five senses in beef cattle, and the training and validation sets were fed into the CFR-YOLO network model for training. The cow face detection model is used to detect the cow’s face from the front, left side, and right side. The cow face part of the picture is detected, and the detected cow face part is feature-extracted using the cow face feature extraction model, which is noted as the front face feature, vector X; the left-side face feature, vector Y; and the right-side face feature, vector Z. Based on the obtained feature vectors, X, Y, and Z, cattle identity is identified using image retrieval techniques, culminating in cattle face detection.

The loss function expression for the training of the cow face feature extraction model is given in Equation (1).(1)$$L_{c}=-\frac{1}{Z}\sum_{i=1}^{Z}x_{i}log\left(R_{i}\right)$$
where $L_{loss}$ represents the loss function; Z represents the total number of cows; $X_{i}$ represents the indicator variable—1 if the current observation is the same as the training sample of the i cow, and 0 if it is different; and $R_{i}$ represents the prediction probability that the current cow is the i cow.

The cow face detection model is specified in Equation (2).(2)$$L_{p}=L_{cls}+L_{box}+L_{mask}$$
where $L_{p}$ is the cow face detection model, $L_{cls}$ is the categorical loss value of the target, $L_{box}$ is the regression loss value of the target, $L_{mask}$ is the target identification loss value, and $L_{cls}$ is calculated by using multicategorical cross-entropy. Therefore, $L_{cls}$ can be calculated using the expression for $L_{loss}$.

$L_{box}$ calculations were made using Equation (3).(3)$$L_{box}=-\frac{1}{N}\sum_{i=1}^{N}R\left(y_{i}-\dot{y_{i}}\right)$$
where N denotes the total number of pixel points in the feature image, $y_{i}$ is the prediction offset vector for the training phase, and $y_{i}^{.}$ is the actual offset vector for the training phase. The R(x) function is the ${smooth}_{L1}$ function. The specific expression is shown in Equation (4).(4)$$R\left(x\right)={smooth}_{L1}\left(x\right)=\left\{\begin{array}{c} 0.5x^{2}if\left|x\right|<1\\ |x|-0.5otherwise \end{array}\right.$$

$L_{mask}$ is determined using the binary classification cross-entropy calculation. The specific expression is shown in Equation (5).(5)$$L_{mask}=-\frac{1}{N}\sum_{i=1}^{N}\left[t_{i}\log\left(x_{i}\right)+\left(1-t_{i}\right)\log\left(1-x_{i}\right)\right]$$
where N denotes the total number of pixel points in the feature image; $t_{i}$ indicates the category, with positive categories being 1 and negative categories being 0; and $x_{i}$ denotes the probability that the sample prediction is positive.

At the time of cattle entering the pen, the facial information of the cattle was first recorded, and the eyes, nose, and mouth were detected using the CFR-YOLO network model of the cattle’s facial key feature points. The center of gravity of the cow’s eyes, nose, and mouth positioning frame and the dimensions of the frame were obtained. The following indicators were calculated as characteristics for identifying beef cattle:

Eye spacing:(6)$$d_{eyespacing}={\left({\left(x_{eye_{l}}-x_{eye_{r}}\right)}^{2}+{\left(y_{eye_{l}}-y_{eye_{r}}\right)}^{2}\right)}^{0.5}$$

Eye average size:(7)$$d_{eyesize}=0.5*\left(\begin{array}{c} {\left({\left(1_{eye_{l}}-1_{eye_{r}}\right)}^{2}+{\left(w_{eye_{l}}-w_{eye_{1}}\right)}^{2}\right)}^{0.5}\\ +{\left({\left(1_{eye_{r}}-1_{eye_{r}}\right)}^{2}+{\left(w_{eye_{r}}-w_{eye_{r}}\right)}^{2}\right)}^{0.5} \end{array}\right)$$

Nose size:(8)$$d_{nosesize}=0.5*\left(\begin{array}{c} {\left({\left(1_{nose_{1}}-1_{ear_{1}}\right)}^{2}+{\left(w_{nose_{1}}-w_{nose_{1}}\right)}^{2}\right)}^{0.5}\\ +{\left({\left(1_{nose_{r}}-1_{nose_{r}}\right)}^{2}+{\left(w_{nose_{r}}-w_{nose_{r}}\right)}^{2}\right)}^{0.5} \end{array}\right)$$

Mouth size:(9)$$d_{nosesize}=0.5*\left(\begin{array}{c} {\left({\left(1_{mouse_{1}}-1_{ear_{1}}\right)}^{2}+{\left(w_{mouse_{1}}-w_{mouse_{1}}\right)}^{2}\right)}^{0.5}\\ +{\left({\left(1_{mouse_{r}}-1_{mouse_{r}}\right)}^{2}+{\left(w_{mouse_{r}}-w_{mouse_{r}}\right)}^{2}\right)}^{0.5} \end{array}\right)$$

The facial video of the cattle was taken at the farm, and the consecutive frames of beef cattle video images were taken at intervals of 0.1 s and input into the deep learning beef cattle. Then, the $n_{eye}$ of beef cattle, $n_{nose}$ of beef cattle, and mouth of beef cattle, $n_{mouse}$, were calculated, and the frames whose detection results satisfied $n_{eye}$ = 2, $n_{nose}$ = 1, and $n_{mouse}$ = 1 were screened out and used as alternative frames.

We then detected the eyes, nose, and mouth of the cattle in the alternative frame images, using the five-senses localization CFR-YOLO for beef cattle, calculating the eye spacing, average eye size, and nose size, respectively, and then comparing them with the above three indexes of all the cattle in the database to calculate the error, which is calculated as in Equation (10).(10)$$\frac{\left|{index}_{2}-{index}_{1}\right|}{{index}_{1}}$$
where ${index}_{2}$ is the feature to be compared, and ${index}_{1}$ is an identifying characteristic for the identity of individual beef cattle in the database. Summing the errors of the above three indicators gives the total error, as shown in Equation (11).(11)$$\cos t=\cos t_{1}+\cos t_{2}+\cos t_{3}$$
where ${cost}_{1}$, ${cost}_{2}$, and ${cost}_{3}$ correspond to the eye spacing, average eye size, and nose size indicators, respectively, to calculate the error; and cost is the total error. All the candidate frames of the cow face detection results are compared with the features of all the cows in the database. The identity corresponding to the minimum total error cost is the result of identifying the individual cow.

### 3.5. Loss Function Improvement

Conventional YOLOv7 uses the CIoU loss function, but CIoU does not take into account the mismatch between the orientation of the prediction frame and the orientation of the real frame. This mismatch of directions can lead to slower and less efficient convergence, and the prediction frames may “wander” during training and ultimately produce poorer models. This paper incorporates the angle between the predicted and actual frames, redefining their distance using angular loss. This approach reduces degrees of freedom, accelerates network convergence, and improves prediction accuracy, and the CIoU loss function in the YOLOv7 algorithm is replaced by the SIoU loss function.

SIoU (Angle, Distance, and Shape Optimization Loss) is an improved IoU loss function that combines angle, distance, and shape losses to further optimize border regression. In the traditional CIoU (Complete Intersection over Union) loss function, the loss function mainly considers the center distance of the border, the aspect ratio, and the overlap area. However, CIoU still has some drawbacks for the situation where the angle and shape are inconsistent, especially when the target attitude changes greatly. To address this, SIoU improved CIoU by introducing the following three losses: First of all, angle loss can effectively punish the angle deviation between the predicted frame and the real target frame by measuring the angle difference between the predicted frame and the real target frame, especially when the object attitude changes, so as to improve the adaptability to the rotating target. Second, distance loss measures the Euclidean distance between the center of the predicted frame and the center of the real frame, which mainly helps the model to accurately locate the target, reduce the displacement error between frames, and ensure the accuracy of the target position. Finally, shape loss optimizes the shape of the frame by calculating the difference in aspect ratio between the predicted frame and the real frame to make it fit the actual shape of the target better, especially when the target shape is irregular or has complex geometric features.

These three loss functions work together in SIoU to optimize the regression of the border, so that the final prediction box is not only accurately positioned and properly sized but also able to adapt to the rotation of and shape changes in the object. Thus, the limitations of CIoU in complex scenes can be overcome, and the accuracy and robustness of border regression can be further improved. Although the CIoU is also optimized in terms of location and size, it neglects the adaptation of angles and shapes, resulting in the failure to achieve the best results in some complex scenes.

SIoU consists of four components: angular loss, Λ; distance loss, Δ; shape loss, Ω; and intersection and merger ratio loss, U. The SIoU parameters are shown schematically in Figure 7.

Angular loss, Λ, is calculated as shown in Equation (12).(12)$$\Lambda =1-2\times \sin^{2}\left(arcsin\left(\frac{C_{h}}{\sigma }\right)-\frac{\pi }{4}\right)$$

Among them, we have the following:(13)$$\frac{C_{h}}{\sigma }=\sin\left(\alpha \right)$$

Based on the angular loss, we define the distance loss, ∆, which is calculated as shown in Equation (14).(14)$$\sum_{t=x,y}\left(1-e^{-\gamma P_{t}}\right)2-e^{-\gamma P_{x}}-e^{-\gamma P_{y}}$$

Among them, we have the following:(15)$$P_{x}=\left(\frac{b_{cx}^{gt}-b_{cx}}{C_{w}}\right),P_{x}=\left(\frac{b_{cy}^{gt}-b_{cy}}{C_{h}}\right),\gamma =2-\Lambda$$

The shape loss, Ω, is defined as shown in Equation (16):(16)$$\Omega =\sum_{t=w,h}{\left(1-e^{-W_{t}}\right)}^{\theta }$$

Among them, we have the following:(17)$$\omega_{w}=\frac{\left|w-w^{gt}\right|}{max\left(w,w^{gt}\right)},\omega =\frac{\left|h-h^{gt}\right|}{max\left(h,h^{gt}\right)}$$
where θ controls the degree of concern for shape loss; and $\left(w,h\right)$ and $\left(h^{gt},w^{gt}\right)$ are the width and height of the predicted and real boxes, respectively. To prevent overemphasis on shape loss and limit the movement of the prediction frame, the θ parameter was restricted to a range from 2 to 6.

The SIoU loss is defined in Equation (18).(18)$$L_{siou}=1-IoU+\frac{∆+\Omega }{2}$$

The model checking pseudo-code of CFR-YOLO is shown in Algorithm 1.

**Algorithm 1.** CFR-YOLO Pseudo-Code
1:INPUT:X//Original feature map2:OUTPUT//z3:Function CFR-YOLO () {4:FOR (I = 1; I ≤ n; i++)5:{6:$L_{box}-\frac{1}{N}\sum_{i=1}^{N}R\left(y_{i}-\dot{y_{i}}\right)$//Total number of pixel points in the feature image7:IF (AP=1) {8:9:

10:$Mn=ə\left(MLP\left(Pool\left(X\right)\right)\right)$//Channel feature weights11:$M_{S}={Ɵ}^{3*3}\left(Pool\left(Y\right)\right)$//Spatial feature weights12:$\;V=\left\{\begin{array}{c} pool{\left(Z^{i}\right)}^{1*1}\to Upsample\\ Prediction\left({Mod}^{x}\right) \end{array}\right.$//Extraction of feature information V13:14:15:$\;B_{k|k+1}=A\left(x1,y1\right)$//Next frame position prediction16:

17:$\;P_{k|k+1}={AP}_{k+1|k+2}A^{T}+Q$//covariance matrix18:$\;\frac{\left|{index}_{2}-{index}_{1}\right|}{{index}_{1}}$//Calculate eye spacing, average eye size, nose size19:$\;cost={cost}_{1}+{cost}_{2}+{cost}_{3}$//Calculate the total error20: ELSE21:  continue22:ENDIF23:}24:ENDFOR25:}26:}

## 4. Methods and Materials

### 4.1. Data Acquisition

This study selected a ranch in Yuanyang County as the site for collecting the cattle dataset. A Canon EOS 5D Mark IV camera was used to capture Simmental cattle at the mature stage in a real breeding scene. The camera uses a full-frame CMOS sensor that provides a high dynamic range and effectively captures the details of the cow’s face. Its advanced Dual Pixel autofocus technology ensures that the focus can be adjusted in real time while the body is moving to ensure the sharpness of the face image. In order to reduce the picture shake in dynamic shooting, the camera has a built-in 5-axis anti-shake system, which effectively improves the shooting stability. In terms of video capture, MP4 format is used for video recording, and the video codec is H.264 format, which has high compression efficiency and small file volume, which is conducive to postprocessing. The video frame rate is set at 30 frames per second (FPS) to ensure smooth video. The resolution is set to 1080p (1920 × 1080 pixels), taking into account the performance of image details and storage requirements. In order to ensure the good generalization and robustness of the model in this study, the collection of datasets followed the following rules: First of all, the cattle video time collected needed to be evenly distributed. The length of each video was controlled to between 150 and 180 s, which can fully capture the regular activities of the cattle while avoiding the creation of redundant data. Second, taking into account the movement of the cattle while on the move, the camera regularly adjusted the shooting angle according to the range of motion of the cattle to ensure that the front, left side, and right side of the face are covered, ensuring that important details are not missed. During the video recording process, the camera starts the recording after waiting for the cow’s face to fully enter the field of view of the camera. When the recording ends, the system automatically stops and saves the video file to improve the shooting efficiency and accuracy. In addition, considering that environmental factors such as illumination and occlusion will affect the detection results of bovine faces, the dataset produced includes not only the bovine face images under the normal environment, but also the bovine face images under a complex background. After frame processing, the video frame is converted into image dataset, the captured video is captured by the Python program, the data are uniformly named, and the cow face images of 80 cows are sorted out in this way. A total of 4973 cow face images were collected, with each cow having 50 to 59 cow face images in varying numbers. The sorted datasets were named FACE1 and FACE2. FACE1 represents images taken under normal conditions. Each image can show the face features of the cow, including the front, left side, and right side, without the face being obscured or affected by changes in lighting. FACE2 represents the bovine face image in a complex environment. Bovine faces in the image are often blocked or affected by different lighting, and there are many cows in the same frame, so the bovine facial features may not be completely visible, and the background is relatively complex. Some of the images captured of the cow face of this ranch are shown in Figure 8 and Figure 9.

The self-built bovine face detection dataset used in this paper was collected in a pasture in Yuanyang County and manually labeled. Each image provides an accurate facial annotation. In the process of data collection and annotation, we strictly followed the code of ethics. All dataset content is for academic research purposes only and does not involve any commercial use.

We are well aware that the use of animal models for data collection in agricultural technologies can raise ethical concerns, especially as the use of animals can involve potential physical and psychological burdens. Therefore, the study strictly followed the standards and regulations related to animal protection and ethics, and all data collection processes were conducted using non-invasive image collection methods to ensure that no physical harm was caused to the animals. We try to minimize the duration of each experiment and control the frequency of the experiments to avoid causing excessive stress or discomfort to the animals. The use of animal testing in agricultural technology research can raise ethically complex issues, so we are committed to strictly enforcing all applicable ethical norms in our research to minimize disturbance to animals through non-invasive and humane means.

#### 4.1.1. Dataset Normalization

In Section 4.1, the image converted from the video is cut, because the different sizes of the cow faces lead to a certain difference in the pixels of the resulting cow face photos. The pixel values of the cow face photos in the cow face dataset produced in the previous section are all between 92 × 196 and 186 × 232. In the convolutional neural network, if the input image is not normalized, it will cause problems, such as difficulty in the extraction of image features, and it will also bring a lot of difficulties in the subsequent classification, so this paper will normalize the size of the cow face images in the dataset to the two kinds of sizes, 224 × 224 and 256 × 256, respectively.

#### 4.1.2. Data Cleansing

The captured video data were sub-framed, and the image data after sub-framing were viewed, and it was found that there are a lot of invalid data, such as incomplete or no cow face in the image. At the same time, the fact that the video was shot artificially may also lead to blurring of the image or a relatively high similarity of the images in the neighboring frames, possibly leading to overfitting during the subsequent training of the model. Therefore, for the invalid data in the above-proposed cow face dataset, the dataset needs to be subjected to a cleaning operation first.

The process of data cleaning is as follows: Firstly, images with incomplete or no cow face are removed using manual screening. The SSIM method is applied to measure the similarity between two images, and those with high similarity are identified. Specifically, if the similarity exceeds a predefined threshold, the two images are considered similar. Subsequently, images ranked lower on the list are filtered out, and the problem of low variance between the cow face datasets can be solved by the SSIM method.

### 4.2. Data Preprocessing

Images due to occlusion by cattle fences in cattle farms, overlapping of cattle faces during cattle movement, weak light adaptation, and poor quality were removed; the dataset was expanded by rotating, flipping, cropping, etc.; and Mosaic data augmentation was used to improve the generalization ability of the detection network.

Mosaic data augmentation enhances the batch size by randomly scaling, cropping, and arranging images to diversify the detection target’s background. The adaptive anchor frame calculation will set the anchor frame in the initial state of network model training, and then it will output a prediction frame, compare the anchor frame with the real frame, calculate the error several times and give feedback, and select the anchor frame with the best adaptability through continuous calculation and compensation, so as to produce the final prediction frame. Adaptive image scaling reduces the black edges added after image scaling by obtaining a smaller scaling factor, which reduces the information redundancy during inference, significantly reduces the amount of computation, and improves the speed of detection. The flowchart of Mosaic enhancement is presented in Figure 10.

In this paper, the data are enhanced by applying horizontal flipping, saturation adjustment, contrast adjustment, luminance enhancement, and luminance weakening methods to the images in the cow face dataset. The enhanced images are displayed in Figure 11.

After performing data enhancement, 9760 cow face detection images were obtained, and the cow face dataset was randomly sampled using. The dataset was split into training, validation, and test sets in a 7:2:1 ratio using the sample method. The details of the division are presented in Table 1.

Table 1 shows the changes in the number of bovine face detection datasets before and after data enhancement. Through the data enhancement method, the number of samples in the training set, verification set, and test set is significantly increased. Before data enhancement, the training set contained 2081 images, the verification set contained 594 images, and the test set contained 297 images. After data enhancement, the sample number of training set increased to 6832, verification set increased to 1952, and test set increased to 1455. Data enhancement not only significantly expands the size of the dataset but also increases the diversity of the data, thereby improving the generalization ability and robustness of the model, and providing a richer sample for the training of the cow face detection task.

#### Data Labeling

After extracting video frames and performing data augmentation, the image dataset for cow face detection is created, along with the corresponding labeled data, i.e., data annotation of the cow face detection image dataset. To carry out data labeling, the Labellmg annotation tool was used to complete the process, and as the next step was to detect the cow face parts in the dataset, the cow face labels were used for calibration. The calibration effect is illustrated in Figure 12. The annotation tool converts image annotations into the YOLO-compatible data format, specifically a txt file, and uses this file and the original image as inputs to the network model. The model is then trained to generate weighted information for subsequent individual identity detection. The labeling should be performed in such a way that, as far as possible, the rectangular box contains only complete cow faces.

### 4.3. Experimental Environment and Parameter Settings

The CFR-YOLO network model proposed in this paper is trained in a deep learning framework; the experimental hardware device has NVIDIA Geforce RTX 3080 Ti as GPU, and Core i7 as CPU; and the integrated development environment is PyCharm environment, based on the programming language Python 3.8.3, based on a 64-bit processor. In this section, the remaining parameters of the network model are standardized. To prevent overfitting, which is common in target detection, the target confidence threshold is set to 0.5, the initial learning rate to 0.001, and the weight decay coefficient to 0.0005. The CFR-YOLO model training parameter settings are shown in Table 2.

Table 2 shows the settings of important parameters in this experiment. All experiments were conducted using the Pytorch 1.2.0 framework. The training process adopted a batch size of 16, momentum of 0.935, and initial learning rate of 0.001, and an attenuation factor of 0.2 was used to adjust the learning rate after each training. A total of 250 EPOchs were trained, and Mosaic was used for data enhancement to improve the diversity of data and the generalization ability of the model. In addition, the optimizer uses the Adam optimizer, which uses a weight attenuation factor of 0.0005 to prevent overfitting. Each time the data are entered, the batch size is set to 20, the number of categories is set to 80, and three boxes are used for detection per prediction box. The step size of the learning-rate decay is set to 1000 to dynamically adjust the learning rate during training. This configuration can improve the training stability and detection accuracy of the model while ensuring the computational efficiency.

In this paper, we evaluate five deep learning networks on the cow face dataset, including YOLOv7, YOLOv5, YOLOv4, SSD, and CFR-YOLO. In order to achieve a fair comparison of the five networks, their main networks and algorithmic hyperparameters were adjusted for the feature requirements of different models, adjusting the depth and structure of the model backbone; selecting appropriate loss functions and optimization methods; and adjusting the experimental settings, such as the amount of training data and batch size. These parameters were reconfigured to maximize the benefits of each model. The reconfigured network parameters are presented in Table 3.

## 5. Results and Discussion

### 5.1. Evaluation Indicators

In this study, accuracy, recall, loss, mean average precision (mAP), and F1 score were employed to evaluate model performance.

Accuracy is a common metric for evaluating classification model performance. It is calculated based on how well the model predictions align with the true labels. A prediction is considered correct if it matches the true label; otherwise, it is deemed incorrect. The calculation formula is provided in Equation (19).

Recall measures the proportion of true-positive examples correctly identified by the model. It reflects the model’s ability to detect all positive samples, with higher values indicating better performance in identifying positive cases. The formula is shown in Equation (20).(19)$$Accuracy=\frac{\left(TP+TN\right)}{\left(TP+TN+FP+FN\right)}\times 100\%$$(20)$$Accuracy=\frac{\left(TP+TN\right)}{\left(TP+TN+FP+FN\right)}\times 100\%$$
where *TP* (true positive) denotes a positively proportional quantity; it is the positive sample for which the model predicts a positive. *FP* (false positive) refers to a negative sample incorrectly predicted as positive by the model. *TN* (true negative) denotes a negative sample correctly predicted as negative. *FN* (false negative) represents a positive sample incorrectly predicted as negative.

The F1 score is the harmonic mean of precision and recall, providing a balanced metric to avoid overemphasis on either precision or recall and offering a comprehnsive assessment of model performance. The formula is shown in Equation (21).(21)$$F1score=\frac{2PrecisionRecall}{Precision+Recall}\times 100\%$$

The loss rate represents the value of the loss function during model training, where the loss function measures the difference between the model’s predicted and true values. The calculation formula is provided in Equation (22).(22)$$Loss=\frac{-1}{n}*\sum \left(y^{*}log\left(y\_hat\right)+{\left(1-y\right)}^{*}log\left(1-y_{hat}\right)\right)$$

Precision measures the proportion of predicted positive cases that are actually positive. The higher precision rate indicates that the model has better detection ability, but the high precision rate does not mean that the model performs well in all cases; it also needs to be combined with other indicators, such as recall rate and F1 value, to comprehensively assess the quality of the model. The accuracy rate is calculated as shown in Equation (23).(23)$$P=\frac{TP}{\left(TP+FP\right)}\times 100\%$$

### 5.2. CBC-YOLOv7 Ablation Experiments

This experiment examines the impact of three improvement methods on the network model. The plotted data are shown in Table 4. We performed ablation experiments in the same environment; added different modules separately; evaluated each module, including the RFB module, CBAM attention mechanism, and CBF module; and compared with the YOLOv7 model, using accuracy, recall, mAP, and F1 as the metrics.

As shown in Table 4, integrating the RFB module into the original YOLOv7 algorithm improves accuracy by 3.84 percentage points and mAP by 1.43 percentage points. After introducing the CBAM attention mechanism to the original YOLOv7 algorithm, the precision, accuracy, recall, and mAP were improved by 4.88%, 4.39%, 2.4%, and 2.84%, respectively. After the introduction of the CBF module to the original YOLOv7 algorithm, the accuracy, recall, and mAP were improved by 2.05%, 0.79%, and 0.37%, respectively. After the simultaneous introduction of RFB module, CBAM attention mechanism, and CBF module to the original YOLOv7 algorithm, the precision rate was increased by 6.92 percentage points. The accuracy increased by 6.23 percentage points, recall by 5.97 percentage points, and mAP by 6.63 percentage points. The precision, accuracy, recall, and mAP of the original YOLOv7 algorithm improved further with the simultaneous integration of the RFB module and the CBAM attention mechanism. And the CBF module’s precision, accuracy, recall, and mAP were higher than those of the original YOLOv7 algorithm with the separate introduction of the RFB module, the CBAM attention mechanism, and the CBF module. This validates the effectiveness of the improved algorithm proposed in this study. The enhanced YOLOv7 algorithm significantly outperforms other algorithms in detection accuracy, precision, recall, and mAP, while maintaining real-time detection capabilities.

This paper demonstrates that the combination of the three modules provides a well-balanced solution, delivering accelerated inference and lightweight deployment performance for the model; qualitatively improves the accuracy of the detection; and brings significant improvements in terms of precision.

### 5.3. Comparative Experiments with Different Networks

In order to verify the superiority of the CFR-YOLO model in accurately recognizing cow faces, five network models were tested on the cow face dataset, namely YOLOv7, YOLOv5, YOLOv4, SSD, and CFR-YOLO, and their accuracies, recalls, loss rates, and mAPs were compared. These five network models are relevant. Firstly, the YOLOv7 network is the original model of CFR-YOLO network. The model’s detection accuracy and speed are enhanced through the efficient aggregation network, reparameterization technique, and dynamic label assignment strategy. Secondly, the YOLOv4 and YOLOv5 networks, the fourth and fifth versions of the YOLO series, use a feature pyramid network to extract features at different scales for target detection. The SSD model is the same single-stage algorithm as YOLOv7. Finally, CFR-YOLO network adds CBAM module, RFB module and CBF module to the improved YOLOv7 network. Therefore, these four network models are selected for comparison experiments.

Table 5 shows the detection accuracy of the five network models on the cow face dataset, and CFR-YOLO performs better than YOLOv7, YOLOv5, YOLOv4, and SSD. CFR-YOLO achieved 98.46% on accuracy, 6.23% higher than YOLOv7, 8.08% higher than YOLOv5, 9.79% higher than YOLOv4, and 12.92% higher than SSD. In terms of recall, CFR-YOLO reached 97.21 percent, 5.97 percent higher than YOLOv7, 7.64 percent higher than YOLOv5, 10.67 percent higher than YOLOv4, and 14.08 percent higher than SSD. CFR-YOLO reduced the loss ratio by 0.07, 0.11, 0.17, and 0.23 compared to YOLOv7, YOLOv5, YOLOv4, and SSD; and it improved the mAP by 6.13%, 8.96%, 10.99%, and 15.25%. These results demonstrate the applicability and accuracy of CFR-YOLO in cattle face feature extraction.

Cow face detection with the addition of feature extraction is a multi-classification task. To compare the convergence efficiency of the CFR-YOLO model with YOLOv7, YOLOv5, YOLOv4, and SSD, this experiment analyzes the changes in training loss values and detection accuracies across training rounds. As shown in the graph in Figure 13, the CFR-YOLO model exhibits faster convergence than the other models for the same detection accuracy. The CFR-YOLO model achieves higher detection accuracy at the same convergence rate. These results highlight the superiority of the CFR-YOLO network model in convergence rate for cow face detection.

Recall rates reflect a model’s capability to identify key information, detect anomalies, and enhance decision accuracy. mAP evaluates the detection algorithm’s performance across various target types, and this experimental analysis shows that CFR-YOLO shows strong learning ability in the initial training phase, achieving faster and smoother convergence, and it produces higher mAP values. The model recall rate and mAP value are shown in Figure 14 and Figure 15.

#### Comparison of Different Network Models in Complex Contexts

We selected single cow face images, multiple cow face images, partially occluded images, and images under varying lighting conditions from the test dataset. These were tested using detection models trained on five algorithms, and the highest MAP values for each model under different conditions were recorded. Table 6 shows that the CFR-YOLO model achieves a MAP value of 96.27% for single cow face detection, while its MAP values for multiple cow face detection and partial occlusion are 93.77% and 87.32%, respectively. The CFR-YOLO model achieves a MAP value of 93.74% under varying lighting conditions, demonstrating its robustness in complex environments. Overall, CFR-YOLO exhibits notable advantages in cow face detection, particularly in complex environments, maintaining consistently high detection accuracy.

Figure 16 illustrates the detection performance of two algorithmic models, CFR-YOLO and YOLOv7, under various cow face image conditions. Since YOLOv7, YOLOv5, YOLOv4, and SSD models perform less effectively than CFR-YOLO, YOLOv7 was chosen for comparison. When a cow face is detected, its location is marked with a red box. If no detection box appears, it indicates that the model failed to identify a cow face. Since there is only one category of beef cow face and the images are very small, the detection-score probabilities are not displayed. As shown in the figure, the YOLOv7 algorithm struggles to detect many small cow faces and exhibits poor localization, often combining two cows into a single detection frame when identifying multiple cow faces. CFR-YOLO outperforms the other four algorithmic models in both cow face classification accuracy and localization precision.

To assess the detection performance of different models across various angles and environments, we selected test dataset images and evaluated the detection accuracy of five models for cow face images under front-face, side-face, and occlusion conditions, as presented in Table 7.

Table 7 reveals that all models perform relatively well in face detection, achieving detection rates generally above 80%. The CFR-YOLO model outperforms all others, achieving a 98.46% detection rate for front faces and maintaining high accuracy for side faces and occlusion cases at 97.53% and 84.06%, respectively. In comparison, YOLOv5 and YOLOv7 perform well, with front face detection rates of 90.38% and 92.23%, but they still fall short of CFR-YOLO. The SSD model recorded the lowest positive face detection rate at 85.54%, highlighting its relative weakness in this task. CFR-YOLO achieves a detection rate of 84.06% under occlusion, demonstrating its robustness in handling such conditions. The accuracy of the other models under occlusion is generally low, with SSD showing the lowest detection rate at 76.31%, reflecting its poor adaptation to occlusion.

### 5.4. Evaluation of the Generalization Effect of CFR-YOLO

To assess the generalization performance of the CFR-YOLO model, the cattle face dataset was divided into a training set and a test set. The model’s generalization effect was evaluated by comparing the accuracy and loss rates between the training and test sets. In cow face detection tasks, cross-subject refers to the fact that the same cow may have different shading, angles, lighting conditions, and other changes in different environments or different positions, which will lead to changes in its facial features. By introducing cross-subjects, these changes can be simulated and processed, so that the model can better adapt to cow face detection in different environments. Cross-view refers to images of the same cow captured from different camera angles or viewpoints. The introduction of cross-views can help the model learn the features in different viewpoints and improve the model’s ability to generalize to different viewpoints. The training and test sets for the cross-subjects in the experiment will include 19,637 and 8951 samples, respectively. The training and test sets for the cross-view include 17,286 and 7683 samples, respectively. As shown in Figure 17, during cross-subject and cross-view training, the model’s accuracy improved with increased training iterations, demonstrating the effectiveness of the proposed method in bovine face detection tasks.

The loss rates corresponding to the training and test sets in cross-subject and cross-view iterative training are shown in Figure 18. As can be seen from the figure, the model loss rate decreases until it is smooth as the number of training sessions increases.

### 5.5. Experimental Analysis of the CFR-YOLO Method Applied to YOLOv8 andYOLOv9

The experiments confirm the feasibility of the proposed improvements to the CFR-YOLO network, demonstrating that the method significantly enhances the model’s performance in target detection tasks. To further demonstrate the generality of the concept, we applied it to YOLOv8 and YOLOv9. In both versions, we observed similar performance improvements, with the experimental results not only supporting the validity of the original hypothesis but also highlighting the method’s adaptability and broad applicability across different YOLO architectures.

The primary evaluation metric during model training is the loss value, which measures the error between the network model’s predicted and actual values. During the experiment, we recorded and saved the loss values at each step and plotted a graph, as shown in Figure 19. The figure shows that YOLOv8 and YOLOv9 initially have higher loss values. However, YOLOv9 exhibits less fluctuation in its loss curve, indicating a more stable training process. Both algorithms complete training after approximately 60,000 iterations. Additionally, the experimental results demonstrate that applying the method results in a clear decreasing trend in the loss curves for YOLOv8 and YOLOv9, with a significant improvement in convergence speed. This demonstrates that the proposed method effectively enhances the training efficiency and performance of various YOLO versions, further confirming its broad applicability in target detection.

In target detection tasks, the MAP value is the standard metric for assessing model detection performance. As shown in Figure 20, we evaluated the two algorithms on the overall test set and plotted MAP value curves for the cow face detection dataset across different iteration counts. The results show that the performance of the two algorithms on the detection dataset corresponds to the decreasing trend of the loss curve in Figure 19, and the smaller the final error (loss) of convergence when the algorithms are trained, the higher the detection accuracy (MAP) of the detection model on the test dataset.

## 6. Conclusions

In this paper, we propose a cow face detection model based on improved YOLOv7. Based on YOLOv7, the CBAM attention mechanism is introduced in the head network, and the RFB module and CBF module are introduced in the backbone network. This model was evaluated by ablation experiments, different model comparison experiments, and generalization effect assessment experiments. The experimental results validated the model’s effectiveness, achieving an accuracy of 98.46%, recall of 97.21%, loss rate of 0.18, and mAP of 96.27% in various model comparison experiments. In addition, we encountered some problems during the experiment, including dataset bias and computation overload. In the task of bovine face detection, there are some biases in the training dataset used; for instance, especially in some specific types of bovine face images, the data samples are scarce. This may result in poor generalization ability of the model for a small number of samples, thus affecting the overall detection accuracy. To solve this problem, we employ data enhancement techniques and try to add more diverse and intrinsically balanced datasets. When dealing with large-scale image data, the computational load of model training and inference is relatively large, which leads to the problem of computation overload. In order to improve the efficiency and reduce the consumption of computing resources, we optimized the model, but after introducing the attention mechanism and various modules, the number of parameters in the original model will be increased, so our subsequent work will focus on the study of model lightweight.

## Figures and Tables

**Figure 1 sensors-25-01084-f001:**
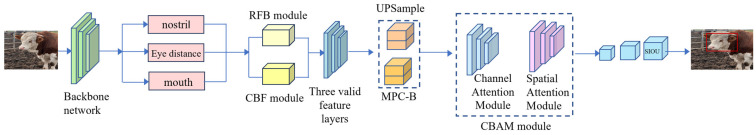
System architecture diagram.

**Figure 2 sensors-25-01084-f002:**
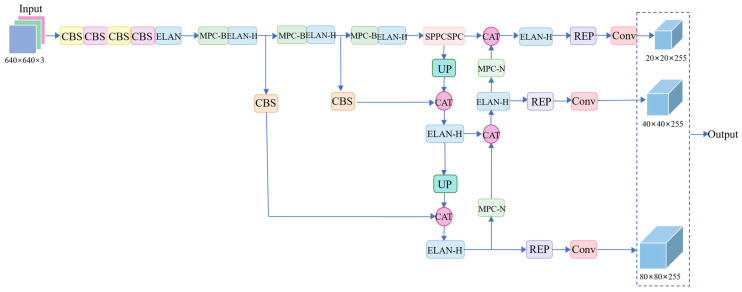
YOLOv7 model structure diagram.

**Figure 3 sensors-25-01084-f003:**
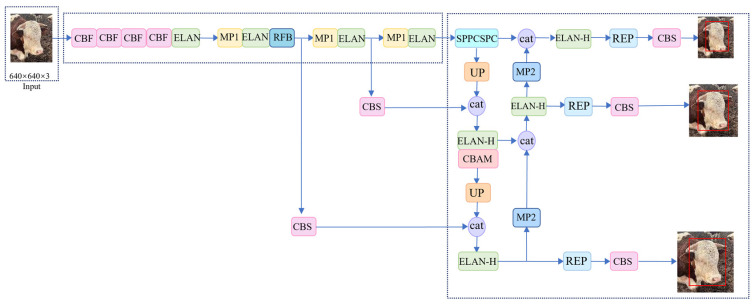
Structural diagram of the enhanced YOLOv7 model.

**Figure 4 sensors-25-01084-f004:**
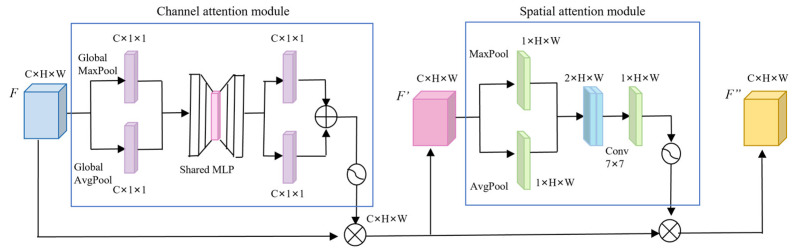
Convoluted attention module. Note: F represents the input feature map; F′ represents the feature map after the channel attention; F″ represents the final feature map; 
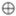
 Shared MLP represents a per-element addition operation; 
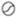
 Global MaxPool represents the Sigmoid activation function operation; 
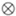
 Globa AvgPool represents the by-element multiplication operation; H and W represent the height and width of the feature maps, respectively; and C is the number of channels.

**Figure 5 sensors-25-01084-f005:**
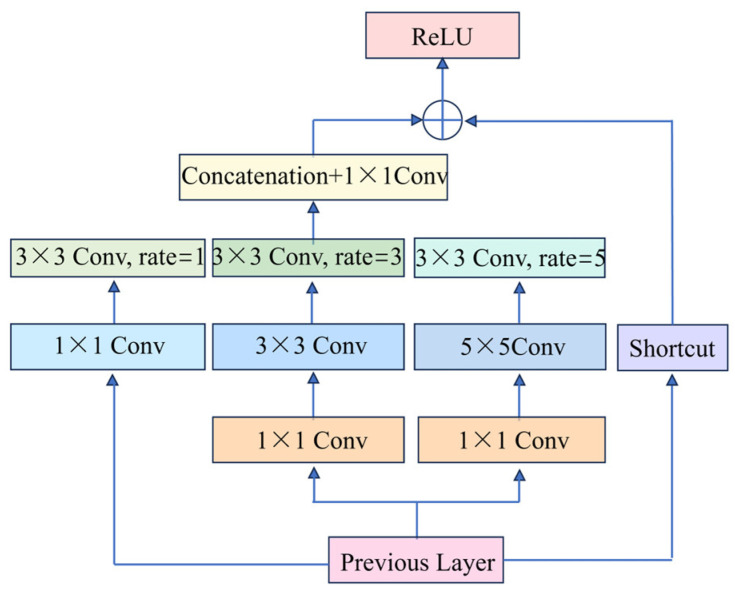
RFB network structure.

**Figure 6 sensors-25-01084-f006:**
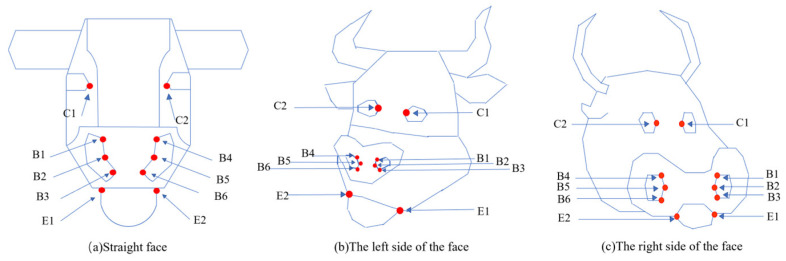
Marking map of cow face feature points.

**Figure 7 sensors-25-01084-f007:**
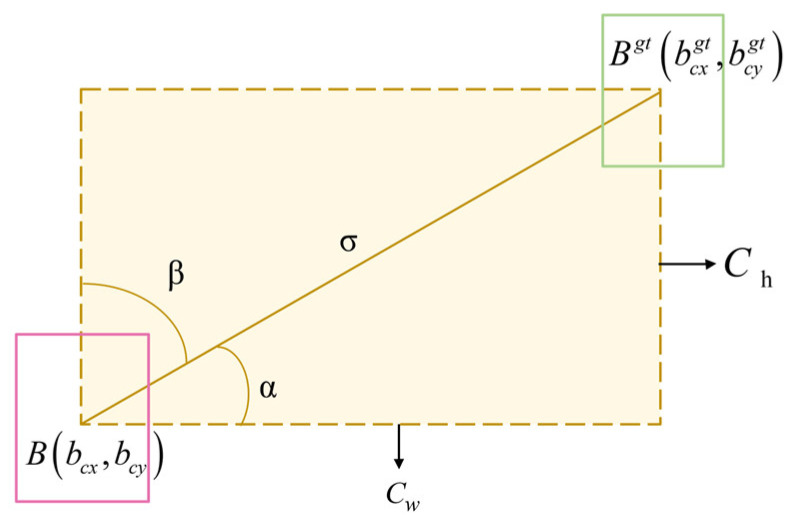
Schematic diagram of SIoU parameters.

**Figure 8 sensors-25-01084-f008:**
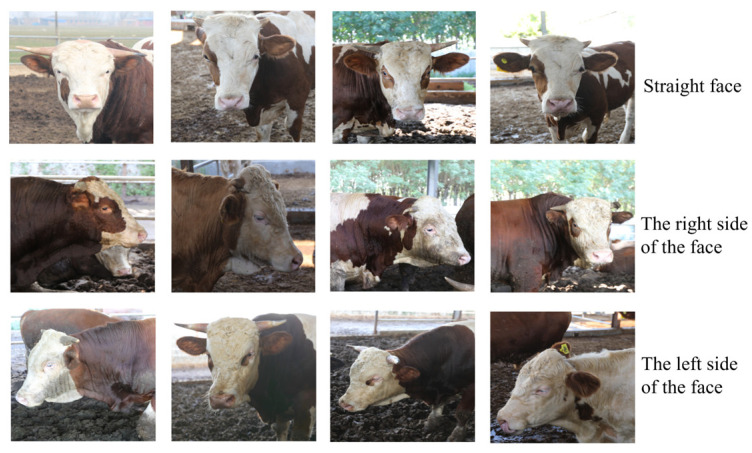
Parts of the cow face, showing left and right face images.

**Figure 9 sensors-25-01084-f009:**
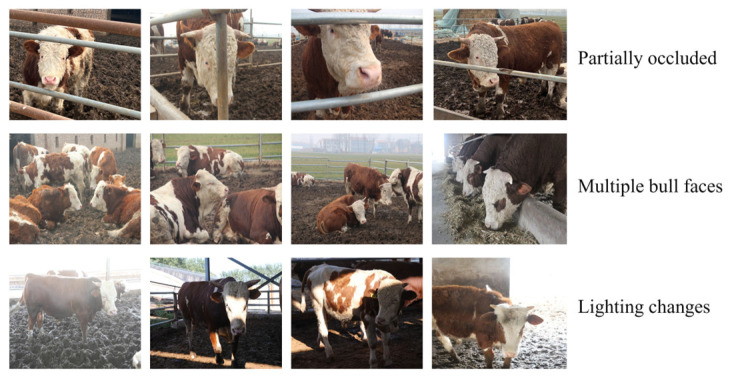
Partial occlusion, light variation, and multiple-cow-face image presentation.

**Figure 10 sensors-25-01084-f010:**
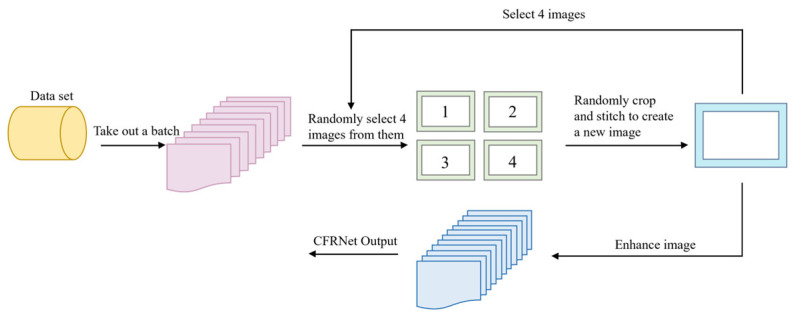
Mosaic data enhancement flowchart.

**Figure 11 sensors-25-01084-f011:**
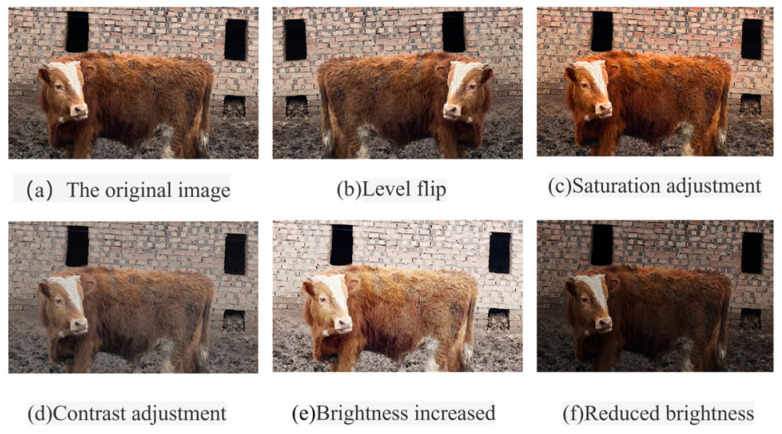
Data enhancement.

**Figure 12 sensors-25-01084-f012:**
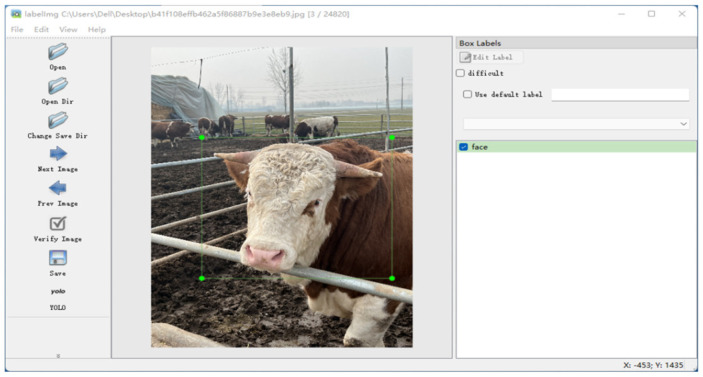
Cow face annotation effect.

**Figure 13 sensors-25-01084-f013:**
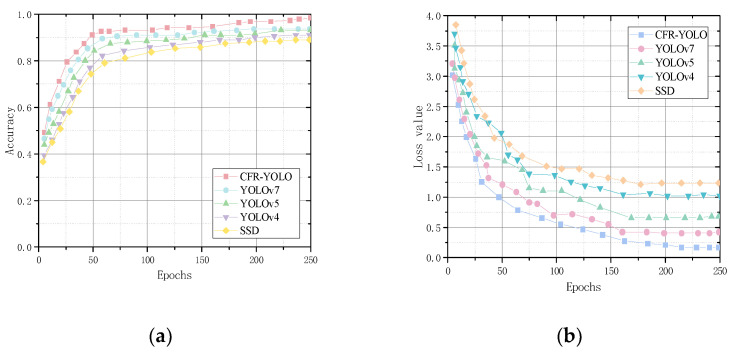
Parison of accuracy and loss rates of different models. (**a**) Accuracy comparison across different models. (**b**) Comparison of loss ratios of different models.

**Figure 14 sensors-25-01084-f014:**
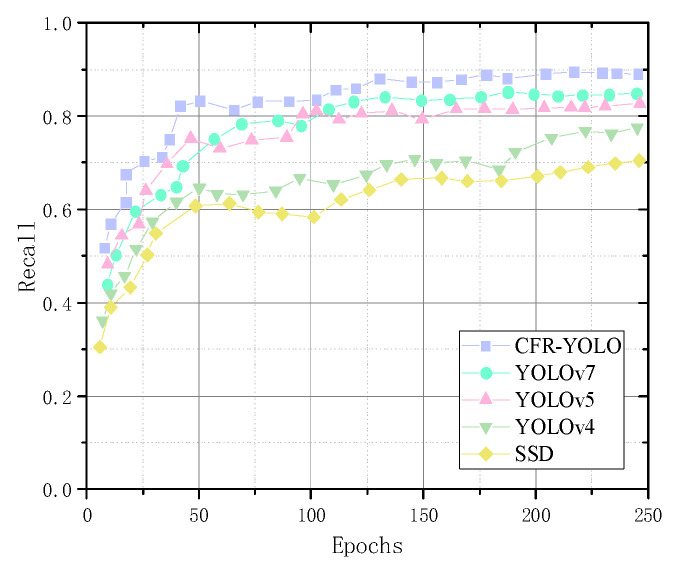
Recall rate comparison across different models.

**Figure 15 sensors-25-01084-f015:**
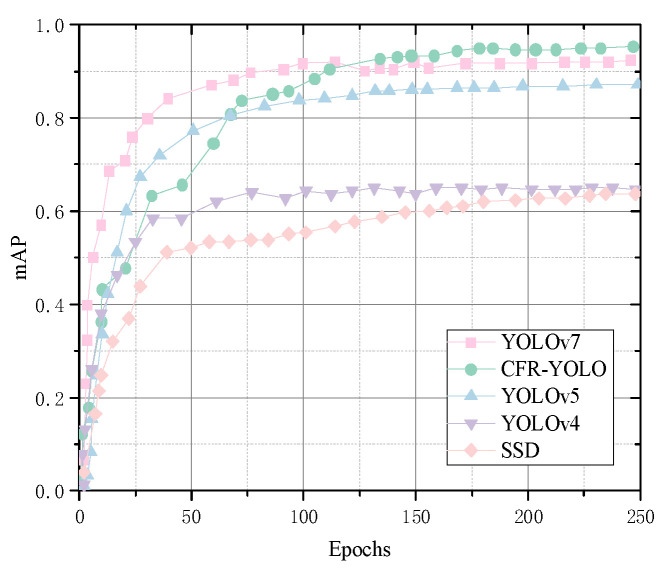
mAP comparison of different models.

**Figure 16 sensors-25-01084-f016:**
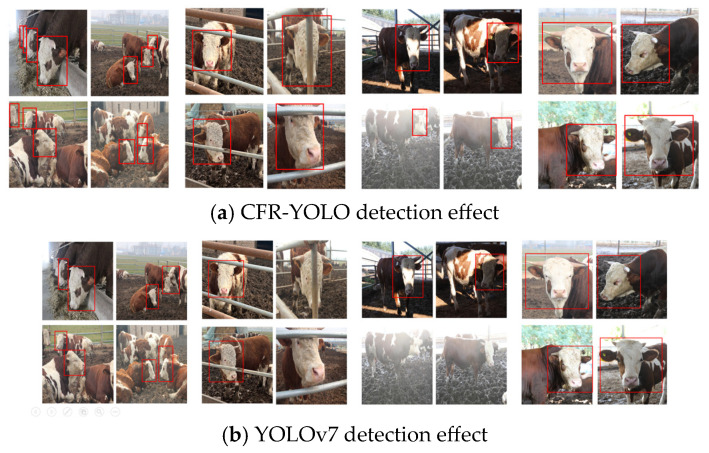
Comparison of the detection effect of the two algorithms.

**Figure 17 sensors-25-01084-f017:**
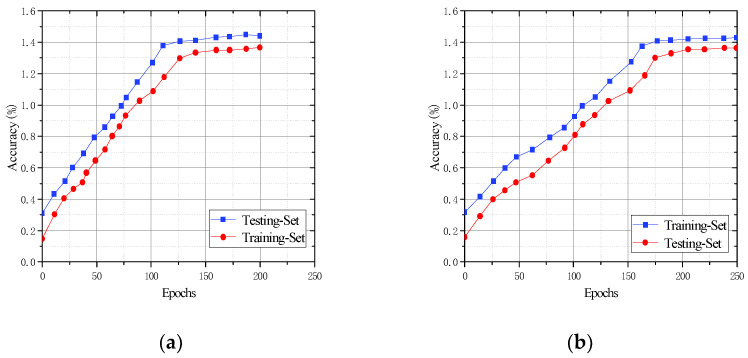
Comparison of accuracy of training set and test set: (**a**) cross-subjects and (**b**) cross-view.

**Figure 18 sensors-25-01084-f018:**
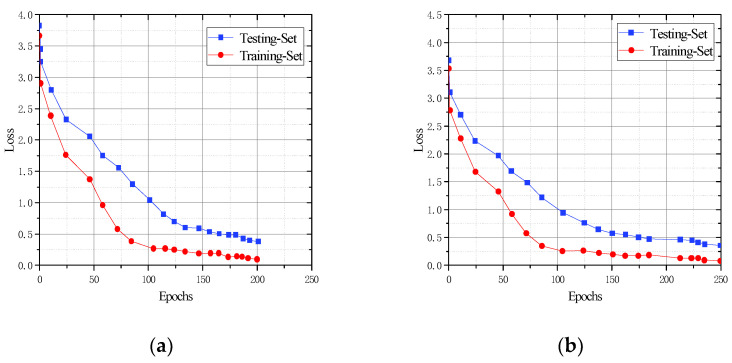
Comparison of loss rate between training set and test set: (**a**) cross-subjects and (**b**) cross-view.

**Figure 19 sensors-25-01084-f019:**
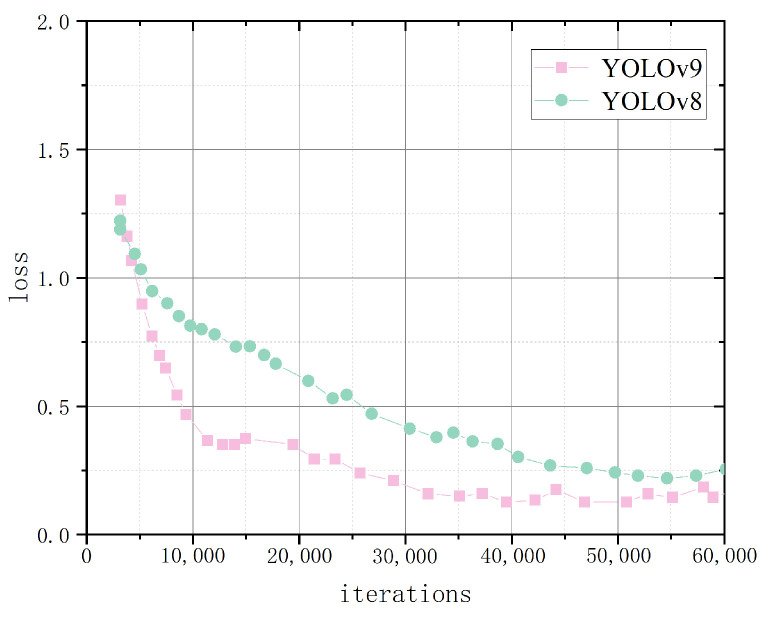
YOLOv8 and YOLOv9 algorithm loss curves.

**Figure 20 sensors-25-01084-f020:**
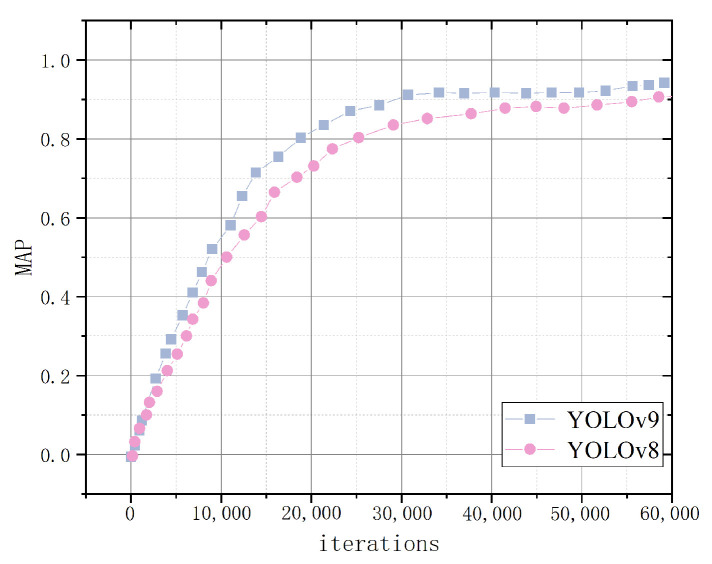
MAP curve of YOLOv8 and YOLOv9 algorithms.

**Table 1 sensors-25-01084-t001:** Division of the dataset of the cow face map.

Cow Face Detection Dataset	Before Data Enhancement	After Data Enhancement
Training set	2081	6832
Validation set	594	1952
Test set	297	1455

**Table 2 sensors-25-01084-t002:** Hyperparameter settings.

Parametric	Be on Duty
Pytorch	1.2.0
Batch size	16
Momentum	0.935
Initial learning rate	0.001
Learning rate decay factor	0.2
Epoch	250
Data enhancement approach	mosaic enhancement
Weight decay factor	0.0005
Quantity per input	10
Number of categories	80
Optimizer	Adam
Number of projected frames/pc	3
Learning rate decay step Batch Size	20 1000

**Table 3 sensors-25-01084-t003:** Reconfigured network parameters.

Network	Parameters	Value
YOLOv7	Conv layer	15
	Max-pooling layer	45
	Learning rate	0.01
	Enter a size	640 × 640
	Weight_decay	0.0005
YOLOv5	Feature map Conv layer Max-pooling layer CRL kernel Feature dimension CBL kernel	128 15 45 3×3 512 5 × 5
YOLOv4	Max-batches Decay Enter a size Activate the function Network structure	20,000 0.0005 640 × 640 Mish CSPDarknet53
SSD	Enter a size Conv layer Conv kernel Feature dimension	512 × 512 30 3 × 3 300
CFR-YOLO	Learning rate Regularization coefficient Batch size Activate the function	0.01 0.0005 16 FReLU

**Table 4 sensors-25-01084-t004:** Results of ablation experiments.

Reticulation	Precision	Accuracy	Recall	Map%	F1%
YOLOv7	90.37	92.23	91.24	90.41	91.73
YOLOv7 + RFB	94.21	96.34	92.52	91.84	94.39
YOLOv7 + CBAM	95.25	96.62	93.64	93.25	95.10
YOLOv7 + CBF	92.53	94.28	92.03	90.78	93.14
YOLOv7 + RFB + CBAM + CBF	97.29	98.46	97.21	97.04	97.83

**Table 5 sensors-25-01084-t005:** Performance comparison of different models on the cow face dataset.

Method	Accuracy %	Recall %	mAP	Loss
SSD	85.54	83.13	81.02	0.41
YOLOv4	88.67	86.54	85.28	0.35
YOLOv5	90.38	89.57	87.31	0.29
YOLOv7	92.23	91.24	90.14	0.25
CFR-YOLO	98.46	97.21	96.27	0.18

**Table 6 sensors-25-01084-t006:** MAP values under different conditions.

Cow Face Detection Model	Single Cow Face (%)	Multiple Cow Faces (%)	Partial Shade (%)	Change in Light (%)
SSD	82.02	78.12	74.52	80.34
YOLOv4	86.28	82.23	78.64	83.83
YOLOv5	88.31	84.51	81.09	86.13
YOLOv7	91.14	88.64	84.22	89.67
CFR-YOLO	96.27	93.77	87.32	93.74

**Table 7 sensors-25-01084-t007:** Detection accuracy in front-face, side-face, and occlusion conditions.

Cow Face Detection Model	Front Face (%)	Side Face (%)	Partial Shade
SSD	85.54	81.42	73.31
YOLOv4	88.67	84.78	75.23
YOLOv5	90.38	87.36	79.52
YOLOv7	92.23	90.23	80.41
CFR-YOLO	98.46	97.53	84.06

## Data Availability

The data used to support this study are available from the corresponding author upon request.

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
