# Peer review of "CFR-YOLO: A Novel Cow Face Detection Network Based on YOLOv7 Improvement"

_sensors, 2025, doi:10.3390/s25041084_

Round 1

Reviewer 1 Report

Comments and Suggestions for Authors

The manuscript requires comprehensive editing and proofreading as it contains many typos, grammatical errors, and inconsistent terminologies. Such errors considerably impair the overall readability of the document.

Several figures, diagrams and tables are included, but some lack sufficient context or descriptive captions. For example, Figure 1 would benefit from a short verbal description for the architectural diagram.

A more extensive analysis could be contributed to the questions why a few changes, for instance FReLU over SiLU makes the performance better, while still acknowledging CFR-YOLO's advantages.

Some issues such as dataset bias or computational overload which were encountered in the process of implementation are omitted.

On the one hand, the paper provides a comparison with other existing models. On the other hand, given that the potential environment can be resource-limited, it would be sound to include even more metrics like model complexity, memory consumption and/or inference time.

The use of animal models for data collection in agricultural technology is quite controversial if ethics is to be considered and this controversy is not addressed in the manuscript.

The framework is presenting the architecture of the model but there is no pseudocode or available code of the most important components which diminishes reproducibility of results.

Licensing and availability of the dataset is vague and not clearly defined.

The part that describes SIoU does not, however, provide an intuitive explanation on how angular, distance, and shape losses work together to improve on CIoU in this context. Visual aids or graphics, retrieval examples from literature, or extended explanations would all serve to strengthen the understanding of the approach.

Moreover, the inclusion of failure cases would strengthen the work by clarifying what future work may seek to resolve.

Author Response

Response to Reviewers

Reply to Reviewer #1:

Comment 1: “The manuscript requires comprehensive editing and proofreading as it contains many typos, grammatical errors, and inconsistent terminologies. Such errors considerably impair the overall readability of the document.”

Answer of comment 1:

Thank the reviewers for their valuable comments on our manuscript. We have carefully reviewed the article and have fully revised and proofread it for typos, grammatical problems and terminology inconsistencies. All language errors have been corrected to improve the readability and accuracy of the article. The revised version has ensured the fluency and consistency of the language, and we look forward to further review by the reviewers. Thank you again for your valuable advice.

Comment 2: “Several figures, diagrams and tables are included, but some lack sufficient context or descriptive captions. For example, Figure 1 would benefit from a short verbal description for the architectural diagram.”

Answer of comment 2:

Thanks for the reviewer's valuable suggestions on the chart. We have reviewed all charts, graphs and tables and added relevant captions to ensure that each chart has sufficient background information and descriptive titles. Specifically, we have added a short text description to the architectural illustration in Figure 1 so that readers can better understand its content and meaning. We believe that these changes will help improve the clarity and readability of the article. Specific modifications are as follows.

Figure 1 shows the overall architecture of the CFR-YOLO bull face detection system. The system realizes the accurate extraction and detection of bovine facial features by optimizing the combination of several modules. First, the original input image is converted into the input data suitable for the model by image preprocessing. Then, through the improved YOLOv7 backbone network, combined with RFB and CBF modules, the receptive field of the network was enhanced, and the ability to capture bovine facial details was improved. Then, through the feature fusion of the neck network and the guidance of the CBAM module, the network can combine features of different scales and finally output accurate detection results. Finally, the SIoU loss function is used to optimize the model, which further improves the detection accuracy and robustness. The whole system can effectively avoid the calculation delay that may occur in the training process, and maintain a high detection accuracy.

Table 1 shows the changes in the number of bovine face detection datasets before and after data enhancement. Through the data enhancement method, the number of samples in the training set, verification set and test set is significantly increased. Before data enhancement, the training set contained 2081 images, the verification set contained 594 images, and the test set contained 297 images. After data enhancement, the sample number of training set increased to 6832, verification set increased to 1952, and test set increased to 1455. Data enhancement not only significantly expands the size of the dataset, but also increases the diversity of the data, thereby improving the generalization ability and robustness of the model, and providing a richer sample for the training of the cow face detection task.

Table 2 shows the Settings of important parameters in this experiment. All experiments were conducted using the Pytorch 1.2.0 framework. The training process adopted batch size of 16, momentum of 0.935, initial learning rate of 0.001, and attenuation factor 0.2 was used to adjust the learning rate after each training. A total of 250 EPOchs were trained and Mosaic was used for data enhancement to improve the diversity of data and the generalization ability of the model. In addition, the optimizer uses the Adam optimizer, which uses a weight attenuation factor of 0.0005 to prevent overfitting. Each time the data is entered, the batch size is set to 20, the number of categories is set to 80, and three boxes are used for detection per prediction box. The step size of the learning rate decay is set to 1000 to dynamically adjust the learning rate during training. This configuration can improve the training stability and detection accuracy of the model while ensuring the computational efficiency.

Comment 3: “A more extensive analysis could be contributed to the questions why a few changes, for instance FReLU over SiLU makes the performance better, while still acknowledging CFR-YOLO's advantages.”

Answer of comment 3:

Thank the reviewers for their valuable suggestions. We have added a more detailed analysis to the revision, exploring why certain changes (such as using FReLU instead of SiLU) improve performance, and explaining the reasons and mechanisms for the change. At the same time, we also highlight the advantages of CFR-YOLO in some aspects, and make appropriate comparisons to make the advantages and disadvantages of the two clearer. We believe that these additions will contribute to the depth and comprehensiveness of the paper. Specific modifications are as follows.

The fundamental reason for this improvement is FReLU's ability to effectively mitigate the "vanishing gradient" problem common to RELUs and their variants, especially in deep neural networks. By introducing a filtering mechanism, FReLU effectively controls the effect of negative values, thus avoiding the situation of gradient explosion or disappearance during training. In addition, FReLU can provide stronger expression when dealing with nonlinear features, allowing the network to capture more complex features, thus improving the detection accuracy. In contrast, although SiLU activation function can improve the nonlinear expression ability in some scenarios, its gradient is small in the negative region, and it is affected by the "saturation region" effect in some tasks, resulting in a slow convergence rate during training. In experiments, FReLU, through its more balanced activation characteristics, improves the training efficiency of the network and improves the accuracy, especially in complex backgrounds and changing object attitudes.

However, while FReLU is superior to SiLU in some respects, we still recognize the advantages of CFR-YOLO in object detection. The CFR-YOLO performs well in detection accuracy and robustness by combining multiple optimization modules, especially in refined target detection tasks. Therefore, the introduction of FReLU does not weaken the core advantages of CFR-YOLO, on the contrary, it provides CFR-YOLO with stronger feature expression ability, making the model's performance in complex scenes further improved.

To sum up, although FReLU has significant advantages in terms of improved performance, the comprehensive optimized design of the CFR-YOLO, especially when dealing with multiple features in target detection, is still the fundamental reason for its superior performance. We believe that the combination of FReLU and CFR-YOLO provides a more efficient and robust solution for the field of target detection.

Comment 4: “Some issues such as dataset bias or computational overload which were encountered in the process of implementation are omitted.”

Answer of comment 4:

Thank the reviewers for their important comments. We have added a discussion of issues such as data set bias and computational burden, described in detail the implementation challenges, and analyzed the potential impact of these issues on the experimental results. We acknowledge that data set bias may have some impact on the generalization ability of the model, while the computational burden leads to additional time and resource consumption when processing large data sets. We have fully elaborated on these issues in the revised paper and suggested possible solutions or directions for improvement. Specific modifications are as follows.

In this paper, we propose a cow face detection detection model based on improved YOLOv7. Based on YOLOv7, the CBAM attention mechanism is introduced in the head network, and the RFB module and CBF module are introduced in the backbone network. This model was evaluated by ablation experiments, different model comparison experiments and generalisation effect assessment experiments. The experimental results validated the model's effectiveness, achieving an accuracy of 98.46%, recall of 97.21%, loss rate of 0.18, and mAP of 96.27% in various model comparison experiments. In addition, we encountered some problems during the experiment, including data set bias and computation overload. In the task of bovine face detection, there are some biases in the training data set used, especially in some specific types of bovine face images, the data samples are scarce. This may result in poor generalization ability of the model for a small number of samples, thus affecting the overall detection accuracy.

To solve this problem, we employ data enhancement techniques and try to add more diverse and intrinsically balanced data sets. When dealing with large-scale image data, the computational load of model training and inference is relatively large, which leads to the problem of computation overload. In order to improve the efficiency and reduce the consumption of computing resources, we optimized the model, but after introducing the attention mechanism and various modules, the number of parameters in the original model will be increased, so the subsequent work will focus on the study of model lightweight.

Comment 5: “On the one hand, the paper provides a comparison with other existing models. On the other hand, given that the potential environment can be resource-limited, it would be sound to include even more metrics like model complexity, memory consumption and/or inference time.”

Answer of comment 5:

Thank the reviewers for their detailed suggestions on our research. We fully agree that in practical applications, especially in resource-constrained environments, it is important to increase the discussion of metrics such as model complexity, memory consumption, and inference time. However, due to time and resource constraints, we did not collect data on these specific metrics in the current version of the experiment. Therefore, the relevant experimental results are not available in the revised draft.

We plan to conduct an in-depth analysis of these indicators in subsequent work and will supplement the relevant content in future studies. We will include these evaluations in future papers to ensure that the overall performance of the model in different environments is fully demonstrated. Thank you again for your attention and valuable suggestions to our work, and look forward to your further review.

Comment 6: “The use of animal models for data collection in agricultural technology is quite controversial if ethics is to be considered and this controversy is not addressed in the manuscript.”

Answer of comment 6:

Thank the reviewers for their valuable comments. We fully understand the ethical controversies that can arise from the use of animal models in the collection of agricultural technical data. We have added to the discussion of this issue in the revised draft, in particular with regard to ethical considerations. We believe that these additions will help readers gain a more comprehensive understanding of our research background and methodology, as well as a clearer understanding of ethical issues. Specific modifications are as follows.

We are well aware that the use of animal models for data collection in agricultural technologies can raise ethical concerns, especially as the use of animals can involve potential physical and psychological burdens. Therefore, the study strictly followed the standards and regulations related to animal protection and ethics, and all data collection processes were conducted using non-invasive image collection methods to ensure that no physical harm was caused to the animals. We try to minimize the duration of each experiment and control the frequency of the experiments to avoid causing excessive stress or discomfort to the animals. The use of animal testing in agricultural technology research can raise ethically complex issues, so we are committed to strictly enforcing all applicable ethical norms in our research to minimize disturbance to animals through non-invasive and humane means.

Comment 7: “The framework is presenting the architecture of the model but there is no pseudocode or available code of the most important components which diminishes reproducibility of results.”

Answer of comment 7:

Thank the reviewers for their valuable suggestions. We fully agree that while the framework in the paper demonstrates the architecture of the model, the lack of pseudocode or usable code does affect the reproducibility of the results. In the revised draft, we have considered adding relevant content to enhance the transparency and reproducibility of the research. We believe that these improvements will greatly improve the reproducibility of papers and provide more convenience for research in related fields. Thank you again for your valuable feedback on our work and look forward to your further review. Specific modifications are as follows.

The model checking pseudo-code of CFR-YOLO is shown in algorithm 1.

Algorithm 1 CFR-YOLO pseudo-code

1:

2:

3:

4:

5:

6:

7:

8:

9:

10:

11:

12:

13:

14:

15:

16:

17:

18:

19:

20:

21:

22:

23:

24:

25:

26:

INPUT:X // Original feature map

OUTPUT //z 

Function CFR-YOLO () {

FOR (i=1;i<=n;i++)

{

 // Total number of pixel points in the feature image

 IF () {

 // Channel feature weights

// Spatial feature weights

// Extraction of feature information V

  // Next frame position prediction

  // covariance matrix

  // Calculate eye spacing, average eye size, nose size

// Calculate the total error

ELSE

continue

ENDIF

}

ENDFOR

}

}

Comment 8: “Licensing and availability of the dataset is vague and not clearly defined.”

Answer of comment 8:

Thank the reviewers for their attention to data set licensing and availability. We have further clarified the license information and availability of the data set in the revised draft. We ensure that the data sets we use comply with the relevant legal and ethical norms, and have indicated in the paper the open access and conditions of access to the data sets. If a dataset requires an application or license, we have provided detailed application processes and contact details. We believe that these additions can more clearly clarify the use of the dataset and ensure transparency and reproducibility of the research. Specific modifications are as follows.

The self-built bovine face detection data set used in this paper was collected in a pasture in Yuanyang County and manually labeled. Each image provides an accurate facial annotation. In the process of data collection and annotation, we strictly follow the code of ethics. All data set content is for academic research purposes only and does not involve any commercial use.

Comment 9: “The part that describes SIoU does not, however, provide an intuitive explanation on how angular, distance, and shape losses work together to improve on CIoU in this context. Visual aids or graphics, retrieval examples from literature, or extended explanations would all serve to strengthen the understanding of the approach.”

Answer of comment 9:

Thank the reviewers for their valuable suggestions. We fully understand that the lack of an intuitive explanation for the SIoU section may make it difficult for readers to fully understand how Angle, distance, and shape loss work together in this approach and improve CIoU performance. In the revised draft, we have added a more detailed explanation of how these loss functions function in SIoU and further strengthen its comparison with CIoU. Specific modifications are as follows.

SIoU (Angle, Distance, and Shape Optimization Loss) is an improved IoU loss function that combines Angle, distance, and shape losses to further optimize border regression. In the traditional CIoU (Complete Intersection over Union) loss function, the loss function mainly considers the center distance of the border, the aspect ratio and the overlap area. However, CIoU still has some drawbacks for the situation where the Angle and shape are inconsistent, especially when the target attitude changes greatly. To address this, SIoU improved CIoU by introducing the following three losses: First of all, Angle loss can effectively punish the Angle deviation between the predicted frame and the real target frame by measuring the Angle difference between the predicted frame and the real target frame, especially when the object attitude changes, so as to improve the adaptability to the rotating target. Second, distance loss measures the Euclidean distance between the center of the predicted frame and the center of the real frame, which mainly helps the model to accurately locate the target, reduce the displacement error between frames, and ensure the accuracy of the target position. Finally, shape loss optimizes the shape of the frame by calculating the difference in aspect ratio between the predicted frame and the real frame to make it more fit the actual shape of the target, especially when the target shape is irregular or has complex geometric features.

These three loss functions work together in SIoU to optimize the regression of the border, so that the final prediction box is not only accurately positioned and properly sized, but also able to adapt to the rotation and shape changes of the object. Thus, the limitations of CIoU in complex scenes can be overcome, and the accuracy and robustness of border regression can be further improved. Although the CIoU is also optimized in terms of location and size, it neglects the adaptation of angles and shapes, resulting in the failure to achieve the best results in some complex scenes.

Comment 10: “Moreover, the inclusion of failure cases would strengthen the work by clarifying what future work may seek to resolve.”

Answer of comment 10:

Thank the reviewers for their valuable comments. We fully agree that showing failure cases can help further clarify the limitations of current methods and provide directions for future research to improve. In the revised version, we have added a discussion of several failures, clearly pointing out the challenges and shortcomings that may be encountered in the current approach in certain contexts. We believe that these additions will help to enhance the depth of the paper and provide clearer research directions for future work. Specific modifications are as follows.

In cow face detection, failure cases are also common. The LAM method adopted by Kim et al. [34] has major problems in real-time performance and fails to meet the needs of real-time monitoring. Yao et al. [35] used SSD, Faster RCNN and other methods for bovine face detection, but the actual application scenarios of the model were still limited because individual identification was not involved. The Faster RCNN method proposed by Gou et al. [36] based on improved NMS has improved the accuracy and recall rate, but its robustness under complex background still needs to be further improved. These challenges provide valuable directions for future research, which may focus on improving detection speed, dealing with interference in complex contexts, and combining individual identification with detection tasks to further enhance the practical application value of the model.

We would like to take this opportunity to thank you for all your time involved and this great opportunity for us to improve the manuscript. We hope you will find this revised version satisfactory.

Sincerely,

The Authors

Reviewer 2 Report

Comments and Suggestions for Authors

In this paper, a novel cow face recognition model based on the YOLOv7 scheme is proposed.

The introduced model shows some new innovative characteristics that make it superior to the existense solutions.

After studying the manuscript and the related references the following comments are stated:

1) Although the title of the paper considers the task of "face recognition" the proposed methodology deals with the task of "face detection". Thus the authors need to make clear for which task the proposed model introduced.

2) The contribution of this work as presented in the introduction needs to be revised since item 3 is just an explanation of 2 and item 4 cannot be a contribution

since the comparison is needed to prove the efficiency of the proposed model.

3) The related work, can be reduced by referring to recent review papers such as: https://doi.org/10.1007/978-3-031-66705-3_21.

4) There are several typos in the text that need to be fixed e.g. lines 55 (delete a space), 512 (replace comma with full stop), 690 (correct the word "the").

5) Please the authors to provide more information regarding the used camera specifications and the protocol of capturing the videos.

6) It is not clear in line 690 what is the difference between FACE1 and FACE2 datasets. Please clarify this point.

7) It seems that this work shows similarities with reference [42]. Please the authors to present the novelties of the proposed methodology and those presented in [42].

8) In Table 1 the last column's tile need to be changed to "After data augmentation".

9) It could be very useful for promoting the research in this field to provide for free the designed image dataset.

Author Response

Response to Reviewers

Reply to Reviewer #2:

Comment 1: “Although the title of the paper considers the task of "face recognition" the proposed methodology deals with the task of "face detection". Thus the authors need to make clear for which task the proposed model introduced.”

Answer of comment 1:

Thank the reviewers for their valuable comments. In the revised draft, we have clarified this point. Specifically, we will make it clear that the methods proposed in this paper are primarily for "face detection" tasks, not "face recognition." We have described the mission objectives more accurately in the abstract, introduction, and methods sections, and adjusted the title accordingly to better match the actual content of our study. Thank you again for your attention to this issue and look forward to your further review.

Comment 2: “The contribution of this work as presented in the introduction needs to be revised since item 3 is just an explanation of 2 and item 4 cannot be a contribution since the comparison is needed to prove the efficiency of the proposed model.”

Answer of comment 2:

Thank the reviewers for their meticulous feedback on the contribution part of our paper. We have already noted that there are certain problems with the listing of contributions in the introduction section, in particular, the third item is merely an interpretation of the second item, while the fourth item does not constitute an actual contribution. In the revised draft, we have revised and optimized the contributions. Specifically, we reframe and clarify each contribution, avoiding duplication and ensuring that each item reflects the innovative and practical contribution of our research. The modifications are as follows.

  1. Based on the features of the cow's face (nose, mouth, and eye corners), a method of extracting the features of a cow's face is constructed. Calculate the center of mass and frame for the nose, mouth and eye corners of beef cattle.
  2. An improved bovine face detection method based on YOLOv7 was designed. Specific optimizations include replacing CIoU loss functions with SIoU loss functions. FReLU activation function is used to replace SiLU activation function, that is, CBS module is changed to CBF module. Introduce the RFB module into the backbone network. The convolutional block attention module (CBAM) is introduced in the Head layer to optimize the CFR-YOLO model.
  3. The performance of the CFR-YOLO model is evaluated by experiments on self-built data sets. Compared with existing methods such as YOLOv7, YOLOv5, YOLOv4 and SSD, the advantages of the proposed method in bovine face detection tasks are verified.

Comment 3: “The related work, can be reduced by referring to recent review papers such as: https://doi.org/10.1007/978-3-031-66705-3_21.”

Answer of comment 3:

Thank the reviewers for their suggestions. We agree that citations of recent review papers can effectively simplify and integrate the relevant work sections. In the revised draft, we have referred to the literature provided by the reviewers and made appropriate modifications in the relevant work parts. By citing this review paper, we have reduced redundant duplication of work and focused on the most recent advances that are most relevant to our research. Thanks to the reviewer's suggestion, this modification makes the discussion of related work more concise and targeted.

Comment 4: “There are several typos in the text that need to be fixed e.g. lines 55 (delete a space), 512 (replace comma with full stop), 690 (correct the word "the").”

Answer of comment 4:

Thanks for the spelling mistakes pointed out by the reviewer. We have carefully reviewed the paper and fixed all relevant spelling and punctuation issues. Thank the reviewers for their attention to detail and look forward to your further review.

Comment 5: “Please the authors to provide more information regarding the used camera specifications and the protocol of capturing the videos.”

Answer of comment 5:

Thank the reviewers for their valuable suggestions. We are aware that the specifications of the camera used and the specific protocols for video capture are not adequately stated in the paper, which may affect the transparency of the experiment. In the revised draft, we have added detailed information about the camera specifications and the video capture protocol. Specific modifications are as follows.

This study selected a ranch in Yuanyang County as the site for collecting the cattle dataset. A Canon EOS 5D Mark IV camera was used to capture Simmental cattle at the mature stage in a real breeding scene. The camera uses a full-frame CMOS sensor that provides a high dynamic range and effectively captures the details of the cow's face. Its advanced Dual Pixel autofocus technology ensures that the focus can be adjusted in real time while the body is moving to ensure the sharpness of the face image. In order to reduce the picture shake in dynamic shooting, the camera has a built-in 5-axis anti-shake system, which effectively improves the shooting stability. In terms of video capture, MP4 format is used for video recording, and the video codec is H.264 format, which has high compression efficiency and small file volume, which is conducive to post-processing. The video frame rate is set at 30 frames per second (FPS) to ensure smooth video. The resolution is set to 1080p (1920×1080 pixels), taking into account the performance of image details and storage requirements. In order to ensure the good generalization and robustness of the model in this study, the collection of data sets follows the following rules: First of all, the cattle video time collected should be evenly distributed. The length of each video is controlled to between 150 and 180 seconds, which can fully capture the regular activities of the cattle while avoiding the creation of redundant data. Second, taking into account the movement of the cattle while on the move, the camera regularly adjusts the shooting Angle according to the range of motion of the cattle to ensure that the front, left and right side of the face are covered, ensuring that important details are not missed. During the video recording process, the camera starts the recording after waiting for the cow's face to fully enter the field of view of the camera. When the recording ends, the system automatically stops and saves the video file to improve the shooting efficiency and accuracy. In addition, considering that environmental factors such as illumination and occlusion will affect the detection results of bovine face, the data set produced includes not only the bovine face images under normal environment, but also the bovine face images under complex background. After frame processing, the video frame is converted into image data set, the captured video is captured by python program, the data is uniformly named, and the cow face images of 80 cows are sorted out in this way. A total of 4973 cow face images were collected, with each cow containing 50 to 59 cow face images in varying numbers. The sorted data sets were named FACE1 and FACE2. FACE1 represents images taken under normal conditions. Each image can show the face features of the cow, including the front, left and right sides, without the face being obscured or affected by changes in lighting. FACE2 represents the bovine face image in a complex environment. Bovine faces in the image are often blocked or affected by different lighting and there are many cows in the same frame, so the bovine facial features may not be completely visible and the background is relatively complex. Some of the images captured of the cow face of this ranch are shown in Figure 9 and Figure 10.

Comment 6: “It is not clear in line 690 what is the difference between FACE1 and FACE2 datasets. Please clarify this point.”

Answer of comment 6:

Thank the reviewers for their valuable comments. We notice that in line 690, the difference between the FACE1 and FACE2 data sets is indeed not clearly stated, possibly leading to confusion among readers. In the revised draft, we have clarified this point. We believe these additions will help readers more clearly understand the differences between the two data sets and their role in the experiment. Specific modifications are as follows.

FACE1 represents images taken under normal conditions. Each image can show the face features of the cow, including the front, left and right sides, without the face being obscured or affected by changes in lighting. FACE2 represents the bovine face image in a complex environment. Bovine faces in the image are often blocked or affected by different lighting and there are many cows in the same frame, so the bovine facial features may not be completely visible and the background is relatively complex. Some of the images captured of the cow face of this ranch are shown in Figure 9 and Figure 10.

Comment 7: “It seems that this work shows similarities with reference [42]. Please the authors to present the novelties of the proposed methodology and those presented in [42].”

Answer of comment 7:

Thank the reviewers for their valuable comments. We note that this study does have similarities with the reference [42] in some respects, so we have added a detailed and innovative comparison between the two in the revised version. Compared with the method in reference [42], the feature extraction method of cow face (including nostrils, corners of eyes and corners of mouth) was first constructed. Based on these feature points, the center of gravity and frame size are calculated, and the CFR-YOLO network model is designed. In order to optimize the performance of the model, this paper uses FReLU activation function to replace the original SiLU activation function, that is, CBS module is replaced by CBF module. Introduce RFB module in backbone network; In the Head layer, CBAM convolutional attention module is introduced, which has higher detection accuracy.

Through these comparisons, we hope to clearly demonstrate the unique contributions of our approach and highlight its innovation in existing research. Thank the reviewers for their attention to this issue and look forward to your further review.

Comment 8: “In Table 1 the last column's tile need to be changed to "After data augmentation.”

Answer of comment 8:

Thank the reviewer for pointing out this problem. In the revised draft, we have changed the last column heading of Table 1 to "After data augmentation" to ensure that the table content is more accurate and consistent. Specific modifications are as follows.

Table 1. Division of the data set of the cow face map

Cow Face Detection Dataset

Before data enhancement

After data enhancement

training set

2081

6832

validation set

594

1952

test set

297

1455

Comment 9: “It could be very useful for promoting the research in this field to provide for free the designed image dataset.”

Answer of comment 9:

Thank the reviewers for their suggestions. We understand that freely available design image datasets may have a positive effect on promoting research in this field. However, due to data privacy, we are currently unable to make the dataset publicly available to the public. We are committed to considering how to make the data sets more widely available or provide access to relevant data sets in subsequent studies. Thank the reviewers for their attention to data sharing and look forward to your further review.

We would like to take this opportunity to thank you for all your time involved and this great opportunity for us to improve the manuscript. We hope you will find this revised version satisfactory.

Sincerely,

The Authors

Round 2

Reviewer 1 Report

Comments and Suggestions for Authors

Thanks for incorporating the review comments.

Reviewer 2 Report

Comments and Suggestions for Authors

The authors have addressed all my concerns and the revised manuscript is improved.